# Oceanic memory of tropical cyclones moderates the Kuroshio current

Deyuan Zhang[1,9], Zhanhong Ma [1,9] ✉, Lijing Cheng [2] ✉, Yanluan Lin[3], Fanghua Xu[3], Zhengguang Zhang [4,5], Yunxia Zheng[6], Jianfang Fei [1] & Michael E. Mann [7,8]

Tropical cyclones (TCs) dramatically disturb the upper ocean and leave subsurface temperature anomalies that persist beyond their lifetimes, representing an oceanic "memory" of TC activity. How this long-term memory affects large-scale ocean circulation remains an open question. Here, we use high-resolution (~0.1°) numerical experiments, in combination with observations, to assess TC's impacts on the Kuroshio current. We show that, collectively, Western North Pacific TCs induce subsurface warming to the right of the Kuroshio due to enhanced mixing and downwelling, and cooling along the Kuroshio main axis primarily through upwelling. In the climatological mean, TCs strengthen the upper right flank of the Kuroshio by ~15% while weakening its main axis by ~4% through geostrophic processes, resulting in a net reduction of the Kuroshio's meridional heat transport by $0.02 \pm 0.02$ PW. On seasonal and interannual scales, TC-induced changes are comparable to the background variability of the Kuroshio, highlighting the long-term cumulative impacts of TCs on ocean circulations and climate.

Tropical cyclones (TCs) are transient but intense weather systems that induce strong interactions between the atmosphere and the ocean[1,2]. The immediate response of the upper-ocean thermal structure to the TCs is typically characterized by a cooled surface and a warmed subsurface confined to a region above the thermocline (near 200 m), which are primarily attributed to the vertical mixing induced by the extreme TC winds[3–6]. Vertical advection (upwelling and downwelling) driven by special wind stress curl of TCs extends to the deeper ocean, especially near the TC center, where the upwelling (or Ekman pumping) could compete and overwhelm the warm effects of vertical mixing, leading to a strong cooling that can reach down to 1000 m[5,7–9]. The cooled surface (and mixed layer), namely cold wake, is restored to normal conditions by anomalous heat fluxes typically within one month[10]. The subsurface temperature anomalies, however, stay in the deeper ocean below the mixed layer and are isolated from the atmosphere[5,11–13], becoming the long-term memories of TCs (here "memories" indicate TC-induced anomalies that last longer than its life, such as subsurface temperature anomalies). These long-lasting temperature anomalies have profound impacts on global climate patterns. For example, the ocean heat pump is suggested to account for ~15% of peak global ocean heat transport[14]; the re-emergence of subsurface warm anomalies within the seasonal thermocline may reduce the SST seasonal cycle by about 10%[7]; and these long-lasting subsurface warm anomalies can enhance the subtropical cell by ~3%[11].

Typically, TCs originate over open oceans, move poleward and westward, and reach their peak intensities before approaching the

[1]College of Meteorology and Oceanography, National University of Defense Technology, Changsha, China. [2]State Key Laboratory of Earth System Numerical Modeling and Application, Institute of Atmospheric Physics, Chinese Academy of Sciences, Beijing, China. [3]Department of Earth System Science, Ministry of Education Key Laboratory for Earth System Modeling, Institute for Global Change Studies, Tsinghua University, Beijing, China. [4]Frontiers Science Center for Deep Ocean Multispheres and Earth System (FDOMES) and Key Laboratory of Physical Oceanography, Academy of the Future Ocean, Chongben Honors College, Ocean University of China, Qingdao, China. [5]Laoshan Laboratory, Qingdao, China. [6]Shanghai Typhoon Institute, China Meteorological Administration, and Key Laboratory of Numerical Modeling for Tropical Cyclone, China Meteorological Administration, Shanghai, China. [7]Department of Meteorology, Pennsylvania State University, University Park, PA, USA. [8]Earth and Environmental Systems Institute, Pennsylvania State University, University Park, PA, USA. [9]These authors contributed equally: Deyuan Zhang, Zhanhong Ma. ✉e-mail: mazhanhong17@nudt.edu.cn; chenglij@mail.iap.ac.cn

continents[15–17]. This path leads to the most pronounced ocean responses occurring in the western part of ocean basins, where the strongest ocean currents, known as western boundary currents (WBCs), are also present. WBCs, including the Kuroshio and the Gulf Stream, are characterized by warm temperatures, narrow axes (~100–200 km), and swift velocities (~1 m s$^{-1}$). These currents are typically near-surface-intensified, with current speeds exceeding 0.1 m s$^{-1}$ in the upper 1000 m[18]. They exhibit significant seasonal variability; for instance, the surface Kuroshio displays seasonal current speed fluctuations[19,20] of about 0.05–0.1 m s$^{-1}$ and volume transport variations[21] of ~0.35 Sv. By carrying a large amount of heat, salt, and organisms from low- to high-latitude regions, the WBCs play a crucial role in regulating both regional and global climate[22–24]. A question naturally arises as to how much the WBCs can be affected by TCs.

Previous studies proposed multiple mechanisms in which TCs influence the WBCs with different or even opposite effects. One mechanism is related to TC wind directions, where winds against the current can significantly interrupt the WBCs[25–27] and winds aligned with the flow direction may enhance the transport[28,29]. These wind-driven ageostrophic current changes, including near-inertial oscillations, though highly energetic, only persist for a few days. By contrast, geostrophic current changes associated with the horizontal pressure gradients persist over longer timescales and typically weaken the WBCs[30–32]. For instance, during Hurricane Matthew, mixing processes eroded the sharp horizontal thermal gradients across WBCs, thereby weakening the Gulf Stream for at least 2 weeks due to the reduced pressure gradient by density changes[26,30]. The wind stress curl of TCs can also contribute to the WBC transport by enhancing vertical advection[32]. The resultant negative pressure anomalies throughout the water column may lead to the formation of a sea surface height (SSH) trough over the WBCs, thus generating an anomalous pressure gradient that counteracts the geostrophically balanced flow[33]. A similar phenomenon has been observed for TCs that directly penetrated the Kuroshio[31]. Besides, oceanic mesoscale eddies are proposed to act as reservoirs for the positive vorticity input from TCs, which can accelerate the Kuroshio as these eddies propagate westward and impinge on the current[34]. However, some cases suggest that mesoscale eddies influenced by TCs may also decrease the Kuroshio current[35]. The multiple mechanisms of TCs can influence the WBCs in complex ways. However, these studies either focus on the near-real TC-WBCs interactions or emphasize the role of mesoscale eddies. The cumulative and long-term impacts of TCs on WBCs are still unclear.

The large-scale oceanic memories of TCs, which can persist for months to even years[14,36,37], are possibly a major contributor to the climatological TC impacts on WBCs. However, how the TCs-induced subsurface temperature anomalies impact the WBCs has not yet been examined. A major difficulty is that the role of TCs cannot be easily picked out from real-world observations; the other is the lack of oceanic data with sufficient temporal coverage and resolution, to resolve the WBCs and capture the long-term signals of TCs. Here, we close this gap by conducting high-resolution (~0.1°) global simulations with and without TCs, combined with satellite observations and reanalysis. A thorough investigation of the climate impacts of TCs is implemented by comparing these twin simulations and performing a composite analysis of TC-induced long-term signals from oceanic observations, focusing on the Kuroshio region, where TCs frequently occur. We find that the ocean retains a long-lasting memory of TCs, which can profoundly modulate the Kuroshio current and potentially influence the climate by regulating poleward heat transport.

## Results

### TCs-induced oceanic changes in the Northwestern Pacific

To quantify the TCs' impact on ocean fields, we conduct two simulations, which are performed using the Parallel Ocean Program (POP)[38], the ocean component of Community Earth System Model (CESM)[39],

with each forced by the JRA55 dataset[40]. In one simulation (the FILT experiment), the wind fields of TCs are filtered out, while in the other (the TCWIND experiment), these fields are modified by inserting vortex fields based on statistical fits to observed TC winds[3,41] (Supplementary Fig. 1). Both simulations are performed at a high resolution of ~0.1° with 40 vertical levels in the upper 1000 m, providing an accurate representation of small- and mesoscale processes that are important for simulating upper-ocean responses to TCs[8,11,42]. The validity of the simulated climatology and of the local oceanic responses to TCs is verified by observations (see detailed in Methods).

**TCs' impact on the Kuroshio current.** The difference between TCWIND and FILT experiments reveals a substantial impact of TCs on the Kuroshio current, which is not spatially homogeneous (Fig. 1a). Specifically, TCs decrease the current speed at the main axis of the Kuroshio, while increasing the current on a wider area at the ocean side of the Kuroshio (Fig. 1a). The maximum anomaly of velocity exceeds 12 cm s$^{-1}$, which is ~12% of the climatological mean Kuroshio current speed (~100 cm s$^{-1}$), indicating a significant contribution of TCs to the Kuroshio.

These simulated TCs' impacts can further be verified using the satellite altimetry observations by comparing the composite difference of SSH and surface geostrophic current between the TC-rich and TC-poor years (Fig. 1b), which are defined according to the annual accumulated wind power input of TCs (Supplementary Fig. 2). When TC activity was relatively strong, the surface geostrophic velocity of the Kuroshio was enhanced on the ocean side and slowed on the shelf side, associated with the spatially inhomogeneous SSH gradient anomalies. The SSH east of Taiwan Island was above normal, and the SSHA along the Kuroshio was negative under the influence of TCs. One limitation of our TC-rich vs TC-poor composite approach is that climate variability related to El Niño–Southern Oscillation (ENSO), which influences Pacific TC activity, could masquerade as an apparent Kuroshio current effect. We guard against this not only by our use of linear regression to remove and estimate ENSO-related impacts (see Methods), but also by in addition performing separate composites for El Niño years and La Niña years. The results of these analyses demonstrate that the TCs impacts is robust with respect to ENSO influences (this matter is revisited later).

Another approach to isolate TCs impacts from background variability involves comparing the SSHA changes across the Kuroshio before and after the passage of TCs[36] (see details in Methods). Taking the results of section 7 for example, 163 cases approaching the section (the TC center within 500 km of the section center) during 1993–2016 are included in the composite analysis (Supplementary Fig. 3a; 500 km is selected as the threshold because this is the radius within which TCs cause significant ocean response[6,36] and the results are not sensitive to the choice of distance threshold as illustrated in Supplementary Fig. 4). The changes of SSHA are averaged over a quasi-steady period between 120 and 180 days to obtain the long-term change in SSH across the Kuroshio induced by TCs (Fig. 1c). This approach is used for satellite data, the HYCOM reanalysis, and the TCWIND simulation. All three cases show that negative SSHAs were found across the Kuroshio after TC passage and reached their maximum on the offshore side, leading to a negative SSHA gradient at the main axis and a positive SSHA gradient on the ocean side, similar to the pattern shown in Fig. 1b.

The impact of TCs on the Kuroshio is not only limited to the surface but also extends to deeper layers, as indicated by the vertical structure of velocity anomaly (Fig. 1d). The simulated velocity anomaly induced by TCs is mainly positive in the top 200 m, indicating that the TCs generally led to an acceleration of the Kuroshio in the upper layer, consistent with ref. 34. However, the negative anomalies become dominant as the depth increases and are maximized at ~200–300 m. HYCOM reanalysis and the composite analyses all verify the simulated results (Fig. 1d and Supplementary Fig. 5), demonstrating that the

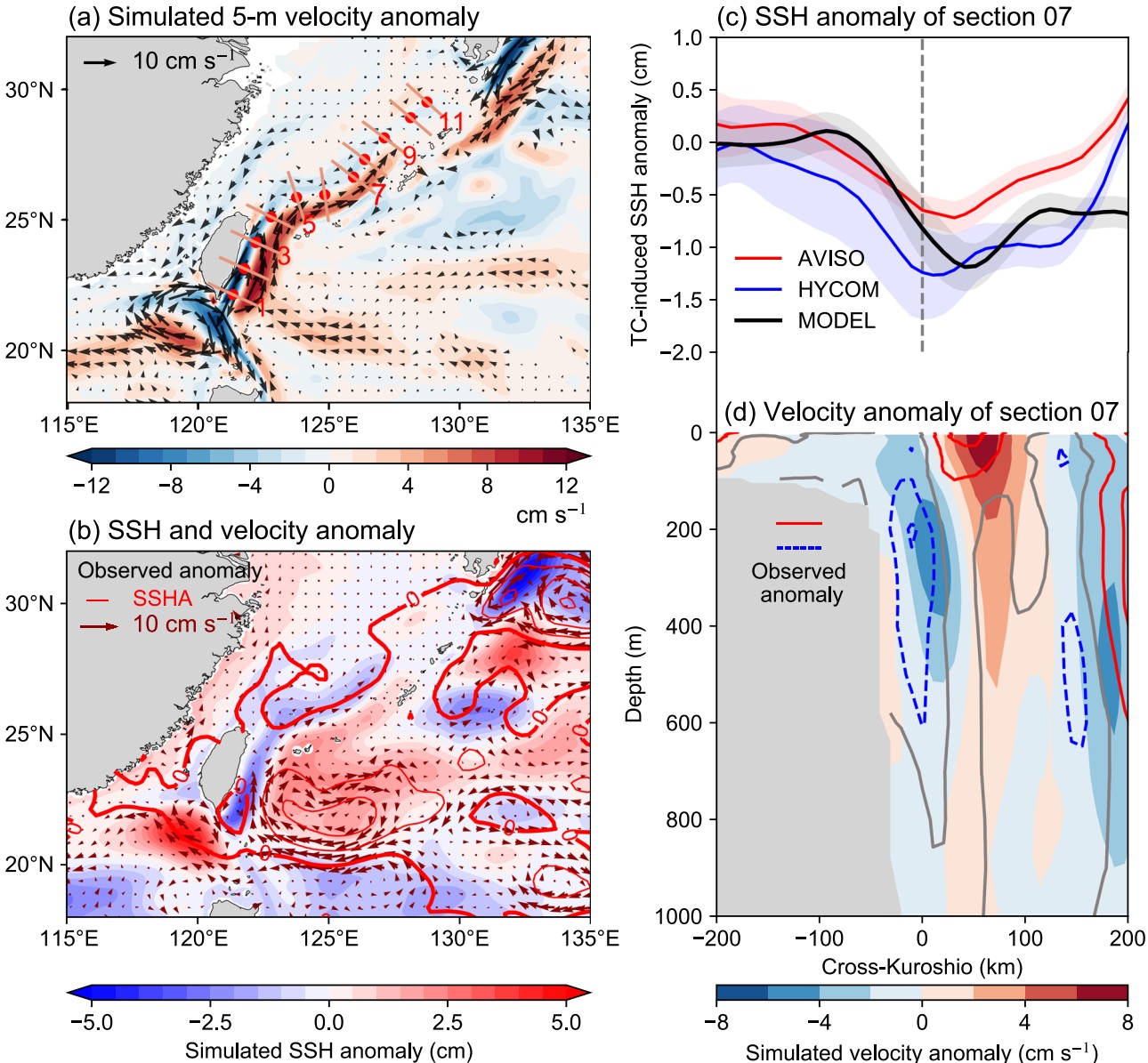

**Fig. 1 | Responses of the sea surface height (SSH) and velocity field to tropical cyclones (TCs) in the Kuroshio region from observations and simulations.** **a** Simulated long-term mean (1993–2016) anomalies of 5 m velocity field (color shading and vectors, cm s⁻¹). The location and length of 11 sections are indicated by red dots and red solid lines, respectively. **b** The mean SSH and surface geostrophic velocity anomalies from observations (contours and vectors) and simulations (color shading). The observed anomalies are calculated as the difference between TC-rich years and TC-poor years from 1993 to 2019. The linear trend has been removed. **c** Tracked mean SSH changes across section 7 over a period of 120 to 180 days after TC passage relative to the pre-storm state (days −30 to −3). Shading indicates standard errors, which are calculated as the standard deviation divided by the square root of the sample size. The black, red, and blue curves are obtained

from the TCWIND experiment, AVISO, and HYCOM reanalysis, respectively. **d** Vertical structure of TC-induced velocity changes across section 7 from the two simulations (color shaded) and HYCOM reanalysis (contours at intervals of 1 cm s⁻¹; the gray contour represents no change, while red and blue denote increases and decreases in velocity, respectively). The results of HYCOM are computed as the mean change over 120–180 days following TC passage compared to the pre-storm state. The composite analysis in (**c**, **d**) includes all TCs within 500 km of the section center from 1993 to 2016 with the linear trend and seasonal cycle removed. All the simulated anomalies are calculated as the long-term mean difference between the TCWIND (with TC embedded) and FILT (with TC removed) experiments. Source data are provided as a Source Data file.

vertical structure is robust. These results indicate that, although the composite velocity anomalies induced by a single TC are quite small (-2 cm s⁻¹), the cumulative effect from multiple sequential TCs is substantial. For example, the maximum velocity anomalies can reach up to 8 cm s⁻¹, which is more than 10% of the climatological state (-80 cm s⁻¹) at that location.

**TCs' impact on the volume transport.** To assess the net effect of these anomlies, the changes in volume transport of the Kuroshio

across the 11 sections from 20° to 30°N are investigated (Fig. 2, the location of these sections is determined according to the maximum velocity and is displayed in Fig. 1a). Although there are some spreads in the magnitudes for the sections, they show similar changes as analyzed before. Integrating the velocity over the upper layer (0–100 m) shows that the volume transport of the Kuroshio increases by about 1.2% while it shows a decrease of about 2.3% when integrated from 100 to 2000 m (Fig. 2a). Although the upper-level velocities are greatest, the net transport is weakened by 1.3% in the

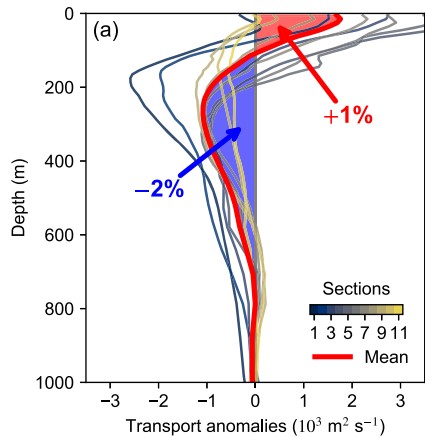
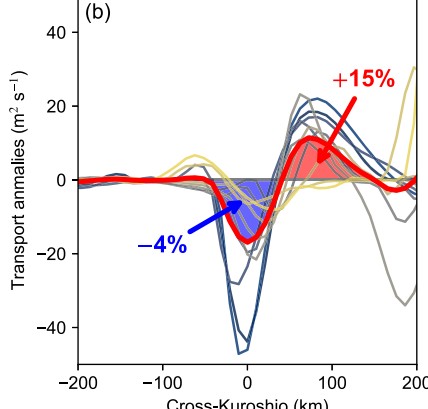

**Fig. 2 | The simulated volume transport responses of the Kuroshio to tropical cyclones (TCs). a** Vertical profile of volume transport anomalies across the 11 sections in Fig. 1, shown by thin lines in different colors. The thick red line represents the average across all sections. The net increases and decreases in volume transport due to TCs are marked in red and blue, respectively. **b** Similar to (**a**), but for the change in vertically integrated transport in the upper 1000 m. Source data are provided as a Source Data file.

upper 2000 m due to the greater depths to which negative velocity anomalies extend. The opposite anomalies between the upper and lower layers indicate that the Kuroshio change due to TC forcing has a significant baroclinic component. Figure 2b shows the transport anomalies across the sections integrated from the surface to 1000 m. The TC-induced local changes across these sections are noticeable: a 4% decline within 100 km around the main axis of the Kuroshio and a 15% increase within 50–150 km on the ocean side, suggesting a significant effect of TCs on the mean current.

## Mechanisms of the TC's impact on Kuroshio

The net weakening of the Kuroshio transport can be understood through the Sverdrup theory[43]. Increased wind stress curl due to wind fields of TCs counteracts the negative subtropical wind stress curl which sets the transport of the western boundary current, finally weakening the vertically-integrated Kuroshio transport in total[44,45]. However, the complex horizontal patterns and obvious baroclinic vertical changes indicate that other physical mechanisms also exist. To understand the underlying processes, we examine the changes in the temperature field under the influence of TCs. The main concept is that the Kuroshio is primarily driven by the strong pressure gradient resulting from the spatially varied density field, which is mainly controlled by the temperature in the subtropics[44,46]. We found that the cumulative effect of TCs on the potential temperature can change the horizontal gradient of the density field around the Kuroshio, leading to changes in the Kuroshio current.

A composite analysis of TC-induced temperature anomalies is conducted in the TCWIND experiment (Supplementary Fig. 6). As previous studies have demonstrated, TCs cause warming in the subsurface and cooling in the deeper ocean[6]. Significant cooling, driven by strong upwelling due to cyclonic wind stress, is most pronounced near the storm center and extends to considerable depths below ~600 m. In contrast, the periphery of the center experiences warming, mainly confined to the upper 200 meters, resulting from downwelling driven by negative wind stress curls and rightward-biased mixing caused by the resonance between wind forcing and mixed layer inertial motion[1,2]. Following the TC passage, these temperature anomalies propagate to the left of the storm track, aligning with the westward movement of the SST and SSH anomalies due to the beta effect[3,36,47]. Notably, we find that the cold anomalies in the deeper ocean, similar to the subsurface warm anomalies, take a long time to dissipate, potentially influencing the large-scale environment.

Figure 3 illustrates the cumulative effect of TCs on the temperature fields around the Kuroshio. The upper layer (50–100 m) of the Kuroshio region experiences predominant warming despite some cooling observed along the main axis of the Kuroshio. The warm anomalies spread along the entire path with the most pronounced warming occurring on the offshore side (Fig. 3a). However, the vertical structure of temperature anomalies (Fig. 3b) indicates that the warm anomalies dissipate rapidly with increasing depth, leaving cold anomalies concentrated at the flow axis (Fig. 3b). The distribution of temperature anomalies is likely influenced by the paths of TCs in the western North Pacific, where TCs typically move westward and then steer northward along the Kuroshio[15,16,48]. Consequently, the cold anomalies are prominent at the main flow axis while the warming flanks the cold center (Fig. 3b), mirroring the local temperature responses (Supplementary Fig. 6a). Besides, the warm anomalies originating from TCs away from the Kuroshio region can also contribute to the current variation. The anomalies induced by TCs can migrate westward with the background circulation and accumulate near the western boundary current, ultimately leading to a warmer anomaly on the offshore side of the Kuroshio. A survey of the thermal response of the Kuroshio after the passage of TCs, using the HYCOM Reanalysis, reveals a similar structure of temperature anomalies, although the warming on the offshore side is less pronounced (Fig. 3d).

Geostrophic velocity anomalies can be calculated from density anomalies according to the thermal wind balance (see Methods). The calculated geostrophic velocity anomalies (red vectors) closely match the model output (black vectors) in both the upper and lower layers (Fig. 3a, b). This similarity indicates the importance of the geostrophic response of the Kuroshio to TCs. The warming concentrating on the ocean side of the Kuroshio (Fig. 3a) accelerates the current along the flanks, with this velocity increase confined to the upper layer, mirroring the distribution of the subsurface warming. In contrast, the cooling extending to ~600 m weakens the Kuroshio throughout its entire depth. As the cooling prevails with increasing depth, the current reduction in the Kuroshio center gradually becomes dominant as discussed above. This phenomenon demonstrates the crucial role of thermal responses in regulating the velocity change of the Kuroshio.

Mesoscale eddies are ubiquitous and dominate the ocean's kinetic energy. Their encounters with TCs have been suggested to notably correlate with the Kuroshio east of Taiwan Island[34,35]. To investigate the possible influence of mesoscale eddies on the Kuroshio, we conduct an additional set of experiments at a lower 1° resolution, which largely eliminates the contributions of mesoscale eddies but retains TC-induced thermal anomalies[42]. The low-

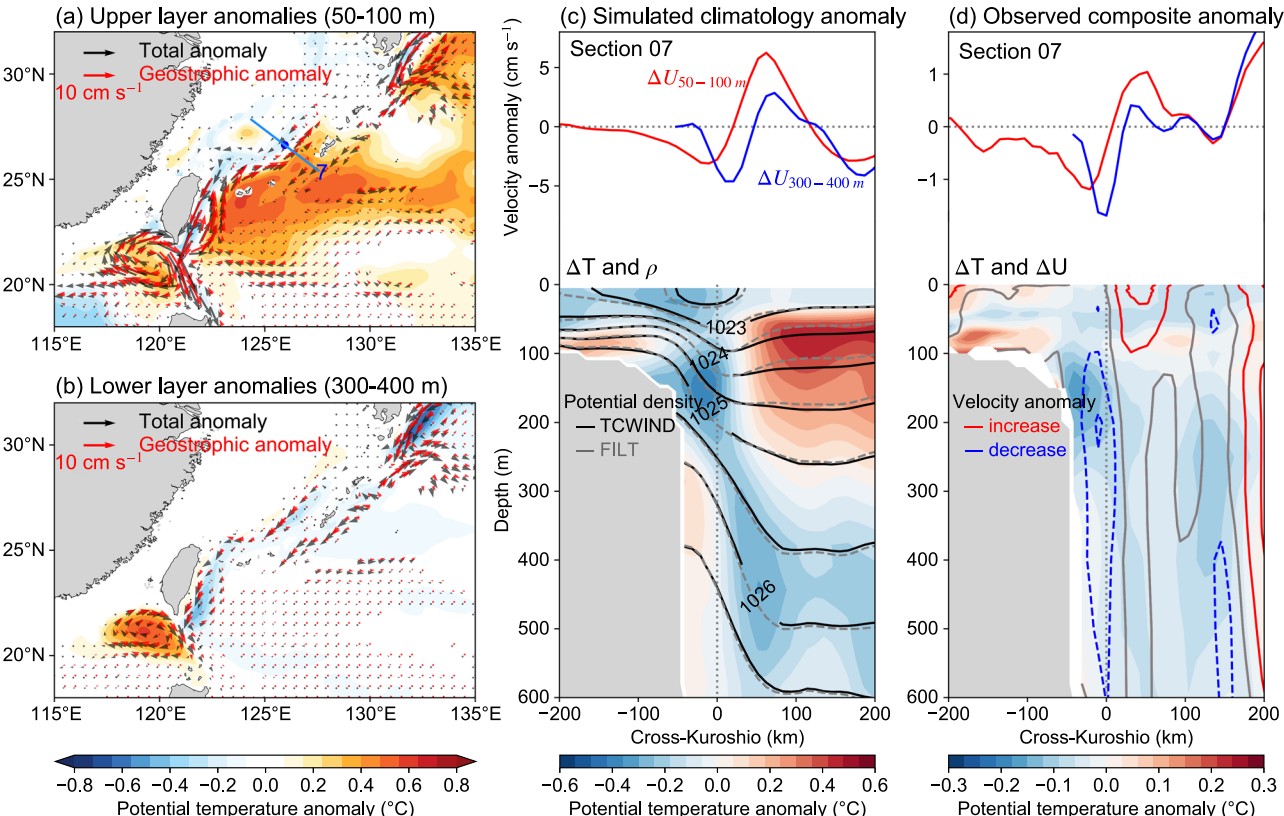

**Fig. 3 | The thermal responses of the Kuroshio to tropical cyclones (TCs) from observations and simulations. a** Simulated upper (50 to 100 m) and **b** lower (300 to 400 m) averaged potential temperature anomalies (color shading) and velocity anomalies (black; vectors) from 1993 to 2016. The red arrows represent geostrophic anomalies calculated from potential temperature anomalies. The blue line indicates the location of the section used in (**c**) and (**d**). **c** Simulated climatology difference of velocity and potential temperature (contours and color shading) across the section. The red and blue lines represent the averaged velocity anomalies over the upper layer and lower layer, respectively. The color shading shows the potential temperature anomalies with the climatology of the TCWIND experiment (with TC embedded) displayed by the contours. **d** Similar to (**c**) but for observed TC-induced change from HYCOM reanalysis. The results are calculated as the mean change over 120–180 days following TC passage compared to the pre-storm state, with the linear trend and seasonal cycle removed, composed of TCs within 500 km of the section center during 1993–2016. Source data are provided as a Source Data file.

resolution model simulates a weaker Kuroshio with a broader flow axis, leading to a less pronounced response to TCs (Fig. 4a). One major difference is the less pronounced cooling of the Kuroshio center (Fig. 4b) compared to the high-resolution experiments, probably due to the reduced Ekman suction at low resolution and consequently less deceleration in deep layers (Fig. 4c). Despite this, the upper layer of the low-resolution model shows a spatial pattern similar to the high-resolution results, characterized by a weakened center and strengthened flanks of the Kuroshio (Fig. 4d). This pattern can also be attributed to the thermal responses induced by TCs, demonstrated as the more prominent warming on the ocean side of the Kuroshio (Fig. 4e). In the high-resolution experiments, mesoscale eddies are explicitly resolved (Supplementary Fig. 7), and their changes after TC passage are also reasonably presented[49] (Supplementary Fig. 8). Nevertheless, both high-resolution and low-resolution models exhibit qualitatively similar spatial distributions, especially in the upper layer, indicating that the Kuroshio current changes could be largely driven by large-scale thermal responses of the ocean to TCs, instead of TCs-eddy interactions.

## Anomalous meridional heat transport due to TCs
Previous studies have shown that TCs contribute to an overall increase in ocean heat content after the cold wake recovery, but estimates vary considerably among various observational and model-based studies (0.14 PW to 1.4 ± 0.7 PW)[6,7,14,50]. Part of this ocean heat pump can be transported equatorward and poleward, essential for maintaining

global heat balance[51]. However, the magnitude of TC-related meridional heat transport also remains uncertain, ranging from 0.035 PW to 0.05 PW from different estimates[3,11,52]. As discussed above, we find that TCs warm the subsurface, but the cooling effect, or cold suction, also cannot be ignored. Moreover, alterations in the thermal structure give rise to modifications in the Kuroshio current. This raises the question of how these two changes will impact the northward heat transport of the WBCs (WHT).

In order to distinguish the contributions of temperature changes and current changes, the overall northward heat transport can be decomposed into three processes[53,54], namely the thermal component driven by temperature changes, the dynamical heat transport change induced by current variations, and the eddy component, i.e. the second-order nonlinear component due to correlated transient departures of current and temperature (See Methods). Figure 5 shows the changes in heat transport within the upper 2000 m in the Kuroshio region under the influence of TCs. The majority of the ocean heat transport is concentrated near the Kuroshio (Fig. 5a), suggesting the pivotal role of WBC in poleward heat transport. Similarly, the heat transport anomalies caused by TCs are also concentrated near the western boundary current (Fig. 5b). The similarity between Fig. 5b, d suggests that changes in the current are likely more significant than thermal responses in regulating heat transport. This implies that when analyzing heat transport changes induced by TCs, it is essential to take into account not only the temperature changes but also their alterations on the velocity field.

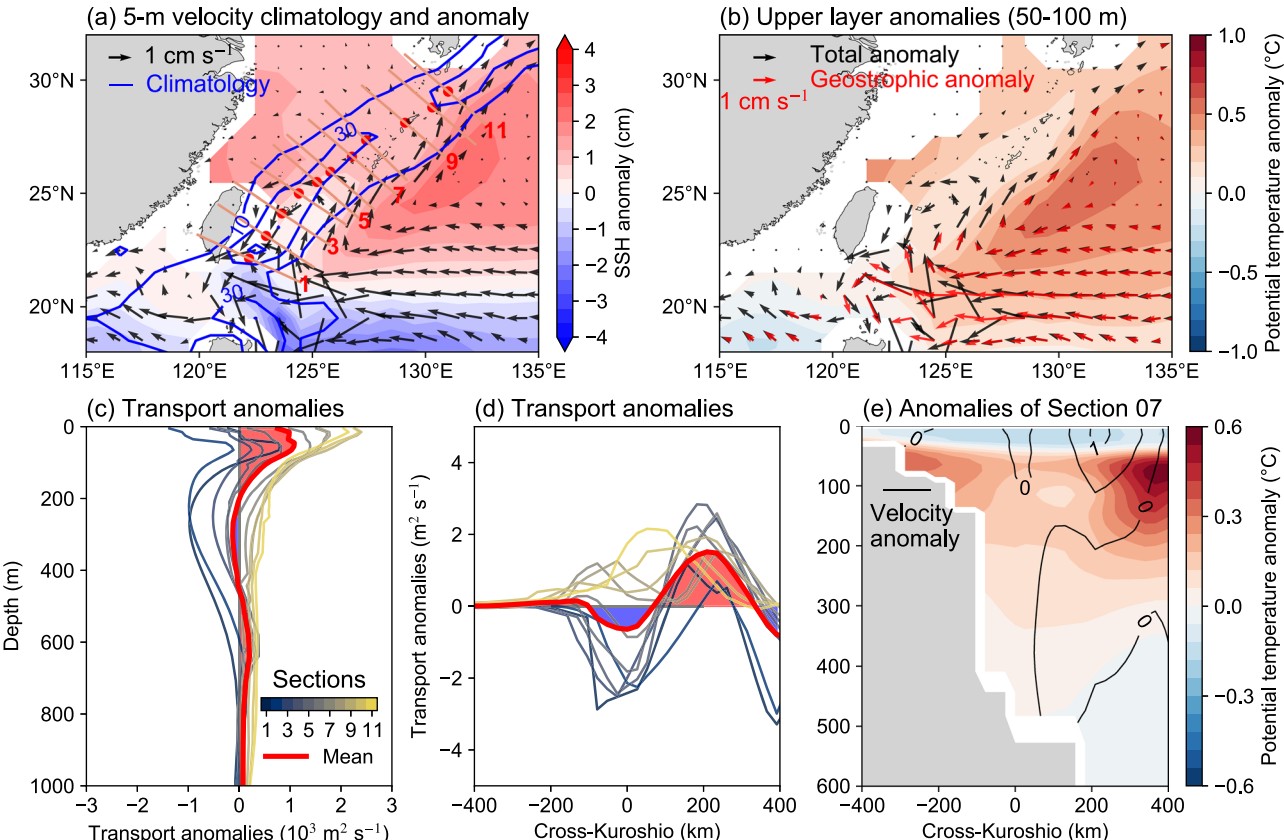

**Fig. 4 | The simulated oceanic responses of the Kuroshio to tropical cyclones (TCs) at a nominal 1° resolution. a** Long-term mean climatology (1993–2016) anomalies of sea surface height (SSH; color shading) and of the velocity averaged over the upper 50 m (vectors). The climatology of the velocity is presented by blue contours at an interval of 20 cm s$^{-1}$. The location and length of 11 sections are denoted by red dots and solid red lines. **b** Climatology anomalies of the potential temperature field over 50 to 100 m. The vectors exhibit the velocity anomalies from the model outputs (black) and calculated geostrophic velocity anomalies (red). **c** Vertical profile of volume transport anomalies across the 11 sections, shown by thin lines in different colors. The thick red line represents the average across all sections. The net increases and decreases in volume transport due to TCs are marked in red and blue, respectively. **d** Similar to (**c**), but for the change in vertically integrated transport in the upper 1000 m. **e** Vertical structure of potential temperature anomalies (shading) and velocity anomalies (contours) across section 7. Source data are provided as a Source Data file.

By defining the area within 5° east of the coastline as the western boundary current region[23,53], Fig. 5e presents the vertical structure of WHT changes along the Kuroshio's main section (20°–30°N), which exhibits similar characteristics to the volume transport change of the Kuroshio (Fig. 2b). An increase in the Kuroshio current within the upper 100 m results in a roughly 10 TW rise of WHT in this layer. However, the cooling and current decrease below 100 m lead to a more significant reduction in heat transport by -27 TW. Thereby, TCs overall reduce the Kuroshio's northward heat transport by about 0.02 ± 0.02 PW (standard error) from 1993 to 2016, of which 58% is attributed to cooling within the Kuroshio, with 40% due to the deceleration of the current. The remaining 2% is associated with the nonlinear interaction term.

## Discussion

Both TCs and WBCs are strong natural processes, and their interactions are important in regulating the regional and global climates. Here, we have demonstrated that long-term thermal responses of oceans to TCs can modulate the Kuroshio current and contribute to northward heat transport. Our findings indicate that not only the shallow subsurface mixing processes but also the deeper-layer cold suction induced by TCs are important modulators of the Kuroshio transport. The spatially uneven warming effect of TCs accelerates the upper flanks of the Kuroshio by about 15%. The 4% weakening of the Kuroshio along its main axis, driven by the cold suction extending into the deeper ocean, is more pronounced, leading to an overall decrease

of volume transport by about 0.3 Sv within 200 km across the Kuroshio. Moreover, our finding emphasizes the necessity of considering current changes when studying the impact of TCs on the ocean heat transport of WBCs. The combined changes in temperature and current result in a decrease of the northward heat transport of the Kuroshio by 0.02 ± 0.02 PW, wherein temperature changes contribute about 58% and flow changes account for 40%. The net weakening of WHT induced by TCs is equivalent to 50% of the projected 100-year change (0.04 ± 0.04 PW) from 1950 to 2050 in a warming climate revealed from the multimodel results of the Coupled Model Intercomparison Project Phase 6 (CMIP6)[53,55], highlighting the important role of TCs in shaping the mean climate state.

In addition to its climatological mean Kuroshio influence, TC impacts are associated with substantial variability in Kuroshio behaviors. TC-induced potential temperature anomalies and current velocity changes are particularly pronounced during TC-active seasons, gradually decaying afterward but maintaining detectable influences on basin-scale circulation (Supplementary Fig. 9). The percentage of TC-related seasonal variation is 33–50% of the seasonal amplitude in potential temperature anomalies (0.5–1 °C versus 1.5–2 °C) and ~50% of the amplitude in current velocity (5–10 versus 10–20 cm s$^{-1}$), underscoring the substantial impact of TCs on seasonal variability of the Kuroshio.

TCs are also associated with substantial inter-annual Kuroshio variability. The inter-annual variation of the Kuroshio itself is closely linked to ENSO through westward-propagated baroclinic Rossby

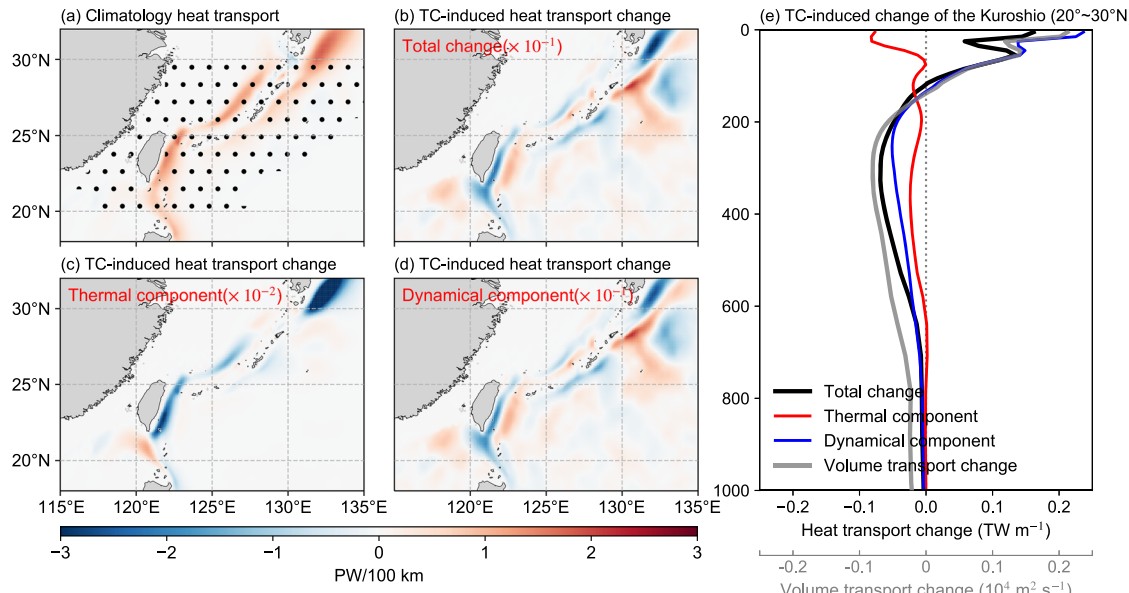

**Fig. 5 | The simulated heat transport responses of the Kuroshio to tropical cyclones (TCs).** Horizontal distribution (PW/100 km) of depth-integrated **a** heat transport climatology for the TCWIND experiment, **b** total heat transport change relative to the FILT experiment (with TC removed), **c** thermal component of heat transport change due to temperature changes, and **d** dynamical component of heat transport change due to current changes. **e** Vertical profiles of the averaged heat transport change of Kuroshio between 20° and 30°N. The total heat transport change (black) is decomposed into thermal component (red) and dynamical component (blue). The northward volume transport change is also noted by the thick gray line. The stippled area in (**a**) is defined as the western boundary region (5° east of the coast) where zonal integration is performed. Source data are provided as a Source Data file.

waves as well as mesoscale eddies in the subtropical countercurrent (STCC) region[56–58]. In addition, however, TC activity in the Northwest Pacific is strongly regulated by ENSO, where TCs are more active, stronger, and southward-shifted during El Niño years[59,60]. Correspondingly, the TC-induced changes in the Kuroshio are stronger during El Niño years than during La Niña years, with a similar spatial domain of influence (Fig. 6a–c). In the TCWIND experiment, a comparison between El Niño and La Niña years reveals that the Kuroshio region exhibits a weakened core and strengthened offshore flow, with a warm anomaly on the east side of Taiwan Island but a cold anomaly on the east side of Luzon Island (Fig. 6e). These features are consistent with reanalysis data (Fig. 6d), but are absent in the FILT experiment (Fig. 6f). Given that the ENSO state is nearly identical in both experiments (Supplementary Fig. 2), we infer that the difference between these two experiments demonstrates the effect of TC activity in causing inter-annual variability in the Kuroshio. These relationships merit further study.

Finally, we suggest that, as climate models advance toward higher resolutions capable of explicitly resolving TCs, it is increasingly important to understand the impact of TCs on the climate itself so that parameterizations of unresolved sub-grid scale processes are physically consistent with the resolved contributions from TCs. In a warming climate, upper-ocean circulation, a significant determinant of the global climate, is projected to change substantially[44,61]. However, the trends in depth-integrated transport of the Kuroshio remain inconclusive and show considerable variability among coupled models[23,53]. Our findings can inform the evaluation of the fidelity of the model estimates, including the potentially important role of TC modulation of Kurishio current and heat transport changes, and may help improve the simulation and prediction of future climate.

## Methods
### Observations
**SST data.** The observational SST climatology from 1993 to 2016 is derived from the Optimum Interpolation SST version 2 (OISSTv2) with a daily temporal resolution and a 0.25° spatial resolution.

**SSH data.** The satellite altimetry data (1993–2016) is used to calculate the SSH and surface geostrophic velocity climatology.

**Subsurface temperature data.** The subsurface (50–100 m) temperature climatology (2005–2016) is calculated from the annual mean gridded IPRC Argo products, which are interpolated in space at 1° resolution from near-real-time Argo profiles.

**TC tracks and winds.** The TC track and maximum sustained wind speed are provided by the International Best Track Archive for Climate Stewardship (IBTrACS) v04 dataset at 6-h intervals.

**TC-induced temperature change.** Based on the information of TCs, the evolution and spatial distribution of the cold wake is calculated by the microwave and infrared data (MW_IR) at 9 km resolution, starting from June 1, 2002 to December 31, 2016. The HYCOM reanalysis (1993–2016) provides subsurface temperature and current with a four-times daily temporal resolution and a 0.08° horizontal resolution and is used to track the responses of Kuroshio after TC passage.

**Oceanic mesoscale eddies.** The satellite altimeter data provides the absolute dynamic topography (ADT) field, which is used to produce daily surface properties and tracks of mesoscale eddies based on the method described in ref. 62.

### Comparison between TC-rich and TC-poor years
The seasonal cycle (subtracting a daily climatology) and linear trend (subtracting a linear trend at each day) were first removed, and then the leading and lagging effects of ENSO were removed via linear regression[63,64]:

$$SSHA_{NOENSO}(t) = SSHA(t) - \sum_{k=-12}^{12} b_k ONI(t-k) \qquad (1)$$

where $t$ is the month in the time series, $SSHA(t)$ is the SSHA without seasonal cycle and linear trend, $SSHA_{NOENSO}$ is SSHA without ENSO

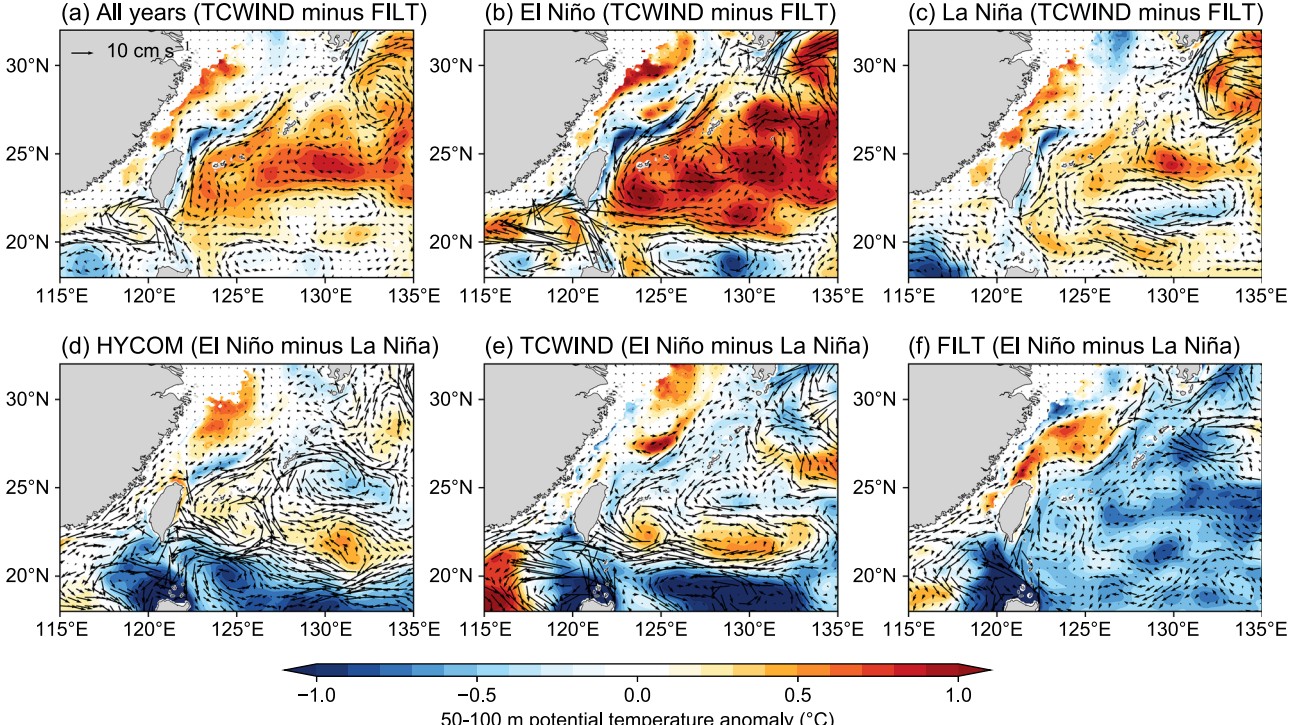

**Fig. 6 | Anomalies in potential temperature (50–100 m; shading) and current velocity (0–50 m; vectors) associated with El Niño–Southern Oscillation (ENSO) during October–December.** Difference between TCWIND (with TC embedded) and FILT (with TC removed) experiments for (**a**) all years (1993–2016), **b** El Niño years, and **c** La Niña years. Differences between El Niño and La Niña years for **d** HYCOM reanalysis, **e** TCWIND experiment, and **f** FILT experiment. Source data are provided as a Source Data file.

effect, $k$ is the lag (negative means lead) month relative to $t$, and $b_k$ is the regression coefficient on the $k_{th}$ ONI. This approach eliminates the significant correlations between SSHA and ONI within leads and lags of up to 12 months (Supplementary Fig. 10), thereby mitigating ENSO-related influences.

After removing linear ENSO effects, we found that a significant negative correlation exists between SSHA and TC intensity along the Kuroshio axis during the TC season (J-A-S). Because of the recovery of TC-induced upper-ocean cooling and a net increase in upper-ocean heat content, a pronounced positive correlation dominates the post-TC season, highlighting the critical role of TC-driven subsurface warming in modulating SSHA variability (Supplementary Fig. 11). These results demonstrate that a comparison of TC-rich and TC-poor years is capable of capturing the impacts of TCs. When calculating the SSHA difference, the annual averaging period was set from July of the current year to June of the following year, accounting for the time lag of the oceanic memory of TC.

## Composite analysis

Composite analysis is employed to isolate the oceanic responses to TCs for both observations and model data. Before the composite analysis, the seasonal cycle and long-term trend are removed for each grid point. The seasonal cycle is represented by a 365-day climatology constructed by averaging all data on each day since 1993. A linear trend of each day is obtained via linear regression. By removing these two components, a dataset without a seasonal cycle and a long-term linear trend is obtained and used to do the composite. The composite was performed from −60 days to +300 days relative to TC passage. The TC-induced oceanic changes were obtained by subtracting the mean pre-storm conditions averaged between day −30 and day −3 before TC passage. The results of composite SSHA change (Supplementary Fig. 3) exhibit pronounced SSHA changes following the TC passage, except in the FILT experiment. Moreover, none of the four datasets displays the

signals of seasonal cycles or trends, indicating that the composite analysis is capable of isolating TC-induced changes. A 120–180-day averaging window was then used to examine the spatial patterns of oceanic responses to TC[36].

## Definition of El Niño and La Niña

El Niño and La Niña years are defined by the Oceanic Niño Index (ONI; Supplementary Fig. 2), which is calculated as a 3-month running mean of SST anomalies in the Niño 3.4 region (5°S–5°N, 120°–170°W). Typically, El Niño (La Niña) events are defined as when ONI is at or above (below) the +0.5 °C (−0.5 °C) anomaly for at least 5 consecutive months. According to the criteria, there are eight El Niño events (1994–1995, 1997–1998, 2002–2003, 2004–2005, 2006–2007, 2009–2010, 2014–2016, 2018–2019) and nine La Niña events (1995–1996, 1998–2001, 2005–2006, 2007–2008, 2008–2009, 2010–2011, 2011–2012, 2016–2017, 2017–2018) from 1993 to 2019.

## Ocean model experiments

**Model configuration.** The model used here is the Parallel Ocean Program version 2 (POP2[38]), which is the ocean component of the Community Earth System Model (CESM[39,65]). An eddy-resolving tripole grid at a nominal 0.1° horizontal resolution is employed, varying from 11 km at the Equator to 2.5 km at high latitudes. It has 62 vertical levels from the surface down to 6000-m depth, with 40 levels in the upper 1000 m. The layer thickness increases from 10 m at the near-surface to 250 m at the deepest level. Compared to the current climate model, typically at around 1° resolution in operation, this finer resolution provides a more accurate presentation of mesoscale eddies, which are reported to significantly modulate the ocean's response to TCs through effects on near-inertial oscillations[42,66], upwelling[5,8], vertical mixing[3,67], and meridional heat transport[11,52] during and after TC passage. Besides, the gross features of western boundary currents are also significantly improved in this horizontal resolution[68]. The model

configuration uses the K-profile parameterization[69] (KPP) as the vertical mixing scheme, which includes processes of shear instability, internal wave breaking, double diffusion, convection, and tidal mixing. Biharmonic horizontal mixing is used for tracers and momentum, and there is no parameterization of eddy-induced mixing. The low-resolution configuration is similar to previous publications[39]. A dipole mesh grid (nominal 1°), with horizontal resolution uniform in the zonal direction (1.125°) but varying in the meridional direction (from 0.27° at the Equator to -0.5° at midlatitudes), is employed. The vertical grid is the same as that in the high-resolution, but has only 60 levels without the two additional levels from 5500 to 6000 m. The vertical viscosity and diffusivity are prescribed by the KPP scheme. The Gent–McWilliams (GM) parameterization[70] is used to approximate the mixing of mesoscale eddies.

**Forcing dataset.** The model is driven by air-sea boundary conditions following the OMIP-2 protocol[55]. The prescribed atmospheric and runoff state is from the JRA55-do dataset[40], providing 3-hourly 10-m air temperature, 10-m humidity, 10-m vector winds, sea level pressure, precipitation, longwave radiation, and shortwave radiation, as well as daily runoff data. The enhanced grid spacing (~50 km) and temporal resolution (3 h) of the JRA55-do dataset provide a more accurate depiction of tropical cyclone characteristics compared to the traditional datasets used in Coordinated Ocean-ice Reference Experiments[71] (COREs). To investigate the net impact of TCs on the ocean, we designed two sets of experiments based on the forcing dataset, one with TC wind information filtered out, referred to as the FILT experiment, and the other with an idealized TC vortex embedded, namely the TCWIND experiment. Besides, another experiment with TC rainfall filtered out is named as NoPrec experiment.

**Filtering and inserting TCs.** To begin with, the 6-hourly TC positions and intensities from the IBTrACS dataset and three-hourly wind fields of the JRA55-do dataset are interpolated to hourly intervals to ensure the temporal evolution of TCs can be properly captured. Following ref. 3, the wind forcing for the FILT experiment $V_{FILT}$ is then obtained by applying an 11-day running mean to the JRA55-do wind forcing within 600 km around each TC position with a linear transition from the filtered to original winds between 600 and 1200 km and expressed as

$$\mathbf{v}_{FILT} = w \times \mathbf{v}_{filtered} + (1 - w) \times \mathbf{v}_{original} \qquad (2)$$

$$w = \begin{cases} 1, & r \leq 600 \\ \frac{1200-r}{600}, & 600 < r \leq 1200 , \\ 0, & r > 1200 \end{cases} \qquad (3)$$

where $r$ represents the distance between a grid point to the TC center. In the original JRA55-do wind fields, the maximum wind speed is weaker than the observed, so the strong winds critical for producing ocean mixing are partially absent. Moreover, the wind structure is rather loose, lacking a compact TC eyewall, which also results in less efficient mixing as the related Ekman suction is probably underestimated. To address this, the ref. 41. idealized wind patterns $V_{vortex}$, which are based on a statistical fit to observed TC winds[72], are embedded into the wind forcing of the FILT experiment to better resolve the spatial structure of TCs. The wind forcing for the TCWIND experiment is obtained as

$$\mathbf{v}_{TCWIND} = w \times \mathbf{v}_{vortex} + \mathbf{v}_{FILT}. \qquad (4)$$

This approach is expected to provide more realistic ocean responses and isolate the impact of TCs on regional and global ocean climates[3,5,7,52].

**Modification of the drag coefficient.** The air-sea fluxes are calculated using the bulk formulation of ref. 73 as a function of the prescribed atmospheric state and the simulated ocean state. However, previous studies[74,75] show that the default drag coefficient $C_D$ tends to overestimate the surface drag under strong winds above 21 m/s. Following previous researches[11,42,52], we adopted the surface drag scheme proposed by ref. 74:

$$z_0 = \begin{cases} \frac{0.0185}{g}(0.001 \times \mathbf{V}_{10m}^2 + 0.028 \times \mathbf{V}_{10m})^2, & \mathbf{V}_{10m} \leq 12.5\, m/s \\ (0.085 \times \mathbf{V}_{10m} - 0.58) \times 10^{-3}, & \mathbf{V}_{10m} > 12.5\, m/s \end{cases} \qquad (5)$$

$$C_D = k^2 (ln\frac{10}{z_0})^{-2} \qquad (6)$$

This modification, together with the inserted TC winds, helps to make the momentum input and ocean mixing under TC conditions more realistic.

**Experiment design.** The model begins with an ocean at rest, initialized with the temperature and salinity from the World Ocean Atlas 2013[76,77] (WOA13) January climatology. It is then spun up for 20 years, forced repeatedly by the original JRA55-do dataset from 1st May 1990 to 30th April 1991. This particular period is chosen because it represents a period that is neutral in terms of major climate modes of variability[78], providing a stable baseline essential for the subsequent experiments. The final state is then branched into two runs from 1991 to 2016: the FILT experiment and the TCWIND experiment. Both experiments are driven by the JRA55-do dataset but with modified 1-h FILT and TCWIND wind forcing, respectively. Similarly, another three experiments are performed using the low-resolution configuration. The difference between the two experiments can then be regarded as the net accumulative influence of TCs on the ocean. All simulated anomalies are calculated as the difference between the TCWIND and FILT experiments.

**Model verification.** The global mean potential temperature reaches a near-equilibrium state after approximately two decades of spin-up (not shown), with no noticeable difference in model drift between the two experiments during the historical simulations. The simulation (in the TCWIND experiment) broadly reproduces the ocean climate observed from 1993 to 2016, including SST, SSH, and subsurface potential temperature (Supplementary Fig. 12). The simulated and observed mean climatology demonstrate broad-scale agreement, with a spatial pattern correlation coefficient (SPCC) of 0.99 for SSH, and with an SPCC of 1.0 for SST and subsurface temperature, indicating that the model realistically captures the main features of the global ocean, such as the positive SSH in the subtropical gyres in the two hemisp and the prominent warm pools in tropical Pacific. For the Kuroshio region, the main studied area, the simulated current in the TCWIND experiment shows good consistency with observations from AVISO and HYCOM reanalysis (Supplementary Fig. 13). These laid the foundation for our subsequent analysis of the changes in the Kuroshio climate state under TC conditions. Moreover, the positions of the Kuroshio axis are estimated based on local maximum velocity according to ref. 79. The results show that the Kuroshio axis is barely shifted as illustrated in Supplementary Fig. 13c. Besides the mean fields, the simulated mesoscale state in the model is also examined using Lagrangian metrics based on identified coherent mesoscale eddies (Supplementary Fig. 7). Despite less frequency, the simulated global mesoscale eddies, in general, agree reasonably well with the observed including the amplitude, radius, and life span.

The ocean responses to TCs are also compared with observations, considering the cold wakes (Supplementary Fig. 14) and interactions

between TCs and mesoscale eddies (Supplementary Fig. 8). The model has reproduced the general distribution, amplitude, and recovery of cold wakes, especially in the Western North Pacific, where TCs are exceptionally strong. The changes in mesoscale eddies after TC passage are also qualitatively similar between simulations and observations, showing strengthened cold eddies and weakened warm eddies. The more prominent responses of cold eddies in the simulations are also seen in other studies, probably attributed to the artificial smoothness in the altimetry-derived eddy properties, as TC-induced effects suddenly occur in -0.5 days[80,81].

**Filtering TC-related precipitation.** The wind, surface heat flux, and rainfall of TCs are all incorporated together in the TCWIND experiment. To investigate precipitation's contribution, an additional experiment is conducted by filtering the TC rainfall. Precipitation rates within a 500 km radius of each TC center were replaced with background values averaged from days −5 to −2 and days +2 to +5 relative to TC passage. A linear transition zone between 500 and 600 km from the TC center was implemented to ensure spatial continuity in the filtered precipitation field. A validation from Typhoon Vongfong demonstrates that this method effectively removes TC-associated precipitation signals both temporally and spatially (Supplementary Fig. 15). A comparison of the model results with and without TC rainfall indicates that the long-term contribution of TC precipitation is marginal relative to that of TC winds to the Kuroshio region (Supplementary Fig. 16).

## Heat transport

This study focuses on the ocean heat transport induced by the Kuroshio. Following ref. 53, a fixed 5° west boundary region from 20° to 30°N is chosen to assess the northward heat transport by the Kuroshio. The western boundary heat transport (WHT) of the Kuroshio is defined as

$$WHT = \rho_0 c_p \int_{\lambda_w}^{\lambda_w + 5°} \int_{-H}^{0} \bar{v}\bar{T} dx dz \qquad (7)$$

where $\rho_0 = 1025 \, kg \, m^{-3}$ is constant reference density; $c_p = 4000 \, J \, (kg \, °C)^{-1}$ is heat capacity; $\lambda_w$ is western boundary, whereas $H$ denotes the depth of ocean; Also, $v$ represents meridional velocity and $T$ is potential temperature. The WHT change ($\Delta$WHT) can be further decomposed[53,54] and expressed as

$$\Delta WHT = \rho_0 c_p \int_{\lambda_w}^{\lambda_w + 5°} \int_{-H}^{0} (\underbrace{\bar{v}_{FILT}\Delta\bar{T}}_{thermal} + \underbrace{\bar{T}_{FILT}\Delta\bar{v}}_{dynamical} + \Delta\bar{v}\Delta\bar{T}) \, dx dz \qquad (8)$$

where $\triangle\bar{v}$ and $\triangle\bar{T}$ are the TC-induced northward velocity and potential temperature anomalies calculated as the difference between the TCWIND experiment and FILT experiment. The equation indicates that the WHT change can be attributed to three processes: (1) thermal heat transport change due to local warming; (2) dynamical heat transport change associated with the transformation of ocean current; (3) the nonlinear contribution of correlated anomalies in both current and temperature. As the calculation is based on the climatology fileds without temporal integration, the nonlinear component is a second-order term that can be generally ignored[53,54]. The anomalies in this study are defined as the difference between the long-term means of the TCWIND experiment and the FILT experiment from 1993 to 2016.

## Geostrophic current anomaly

Assuming a "depth of no motion" $z_{nm}$, typically in the range 1000–2000 m, the geostrophic currents are obtained according to the thermal wind balance. The anomalies of geostrophic currents are then calculated through the density difference between the TCWIND experiment and FILT experiment and expressed as

$$\begin{cases} \Delta u_{g,-h} = -\frac{g}{\rho_0 f}\frac{\partial}{\partial y}\int_{-z_{nm}}^{-h} \Delta\rho(z)dz \\ \Delta v_{g,-h} = \frac{g}{\rho_0 f}\frac{\partial}{\partial x}\int_{-z_{nm}}^{-h} \Delta\rho(z)dz \end{cases} \qquad (9)$$

As the TC-induced temperature and salinity changes are largely confined to the upper 1000 m and the velocity changes at 1000 m are negligible compared to those in the upper layer, we choose 1000 m as the "depth of no motion".

## Statistics and reproducibility

No statistical method was used to predetermine sample size. No data were excluded from the analyses.

## Data availability

The observational SST climatology (OISSTv2) is available at https://www.ncei.noaa.gov/metadata/geoportal/rest/metadata/item/gov.noaa.ncdc:C00844/html). The gridded Argo products can be downloaded from https://apdrc.soest.hawaii.edu/projects/Argo/data/gridded/On_standard_levels/. The satellite altimeter data are provided by Archiving, Validation, and Interpolating of Satellite Oceanographic altimeter data set distributed by the Copernicus Marine and Environment Monitoring Service and can be downloaded from https://data.marine.copernicus.eu/product/SEALEVEL_GLO_PHY_L4_MY_008_047/description. The HYCOM reanalysis dataset is available at https://www.hycom.org/dataserver/gofs-3pt1/reanalysis). The TC information is provided by the International Best Track Archive for Climate Stewardship (IBTrACS) v04 dataset (https://www.ncdc.noaa.gov/ibtracs/). The high-resolution MW_IR SST data is at https://data.remss.com/SST/daily/mw_ir/v05.1/. The ONI data can be downloaded at https://www.ncei.noaa.gov/access/monitoring/enso/sst. Due to the large volume of model outputs, the original modeling data are available upon request (deyuanzhang@nudt.edu.cn). The processed data and codes to reproduce all figures in the article are available at https://doi.org/10.6084/m9.figshare.29432774.v1. Source data are provided with this paper.

## Code availability

The community Earth System Model (CESM) developed by the National Center for Atmospheric Research can be downloaded online at http://www2.cesm.ucar.edu/. Analysis and figure generation were performed using Python.

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

## Acknowledgements

This work is funded by the National Natural Science Foundation of China with Grant 42192552 (J.F.), 42475011 (Z.M.) and 42176019 (F.X.). The simulations were supported by the National Key Scientific and Technological Infrastructure project "Earth System Numerical Simulation Facility" (EarthLab). We thank Dr. Wenchao Chu for assisting with the simulations.

## Author contributions

D.Z. conducted the simulations and performed the analyses and drafted the manuscript; Z.M. and L.C. conceived the project and interpreted the results. M.E.M., Y.L., Z.Z., F.X., Y.Z., and J.F. discussed the results and commented on the manuscript. All the authors contributed to improving the manuscript. D.Z. and Z.M. contributed equally to the work.

## Competing interests

The authors declare no competing interests.
