## [Transparent Peer Review file · Nature Communications]

Oceanic memory of tropical cyclones moderates the Kuroshio current

Corresponding Author: Professor Zhanhong Ma

Version 0:

Reviewer comments:

Reviewer #1

(Remarks to the Author)

This manuscript investigates the influence of “long-term” ocean memory of TCs on the Kuroshio current. The authors conduct two sets of ocean model simulations, forced with and without TC winds, at both eddy-resolving and non-eddy-resolving resolutions, to isolate the impact of TC winds. In addition, they analyze TCs’ impact using satellite observations and HYCOM reanalysis through composite analysis. The authors conclude that Northwestern Pacific TCs can induce ocean temperature anomalies in the Kuroshio region, which modulate the Kuroshio current through geostrophic processes.

This is a very interesting and comprehensive study and it has the potential to make an important contribution to the field. The manuscript is well-written, and the results are generally robust. However, I have several major concerns, particularly related to the methodologies for analyzing reanalysis and satellite observations. I am concerned that these methods do not sufficiently isolate the impact of TCs. Other concerns relate to the role of large-scale modes of climate variability and model drift. I recommend major revision and hope my suggestions can help improve the manuscript. Please see specific comments below:

Major comments:

1. My primary concern is whether the composite analysis of the observational data effectively separates the “ocean memory” of TCs on the Kuroshio current. Given the rich eddy activities and the strong exchange of heat and mass in this region, it is hard to argue the temporal changes are induced by TCs alone. It is somewhat surprising that a composite analysis can still capture TCs’ influence after 120-180 days, as shown in Figure 3 and the Extended Figures 3-5.

For example, in Extended Figure 3, the SSH changes in AVISO and HYCOM are most significant in the first 30 days following the TC passage and have a clear trend of rightward propagation. After 30 days, the SSH shoaling is confined to the section center and is mostly stationary. These are likely two distinct processes: the first involves a short-term response to TCs, and the second is likely driven by longer timescales and may not necessarily be a “memory” of TCs. Moreover, the composite analysis suggests that during the 180 days, only TCs near the section can impact this region and that their impact is confined locally. However, other processes could also affect the boundary current, and TC-induced ocean heat anomalies farther away from the region could be transported to the Kuroshio by ocean currents.

I recommend providing more details on how the seasonal cycles and trends are removed, as well as any post-processing applied to the data. It also would be helpful to extend the composite temporal analysis (i.e., in Extended Figure 3) beyond 180 days to identify if there’s any signal of repeating seasonal cycles or trends. In addition, I would also suggest performing the same composite temporal analysis on the TCWIND experiment and comparing the results with the anomalies calculated as the difference between TCWIND and FILT.

2. Another issue with the observational analysis methodology is the comparison between TC-rich and TC-poor years (Line 148). In Northwestern Pacific, the TC season peaks in late summer to fall, and the annual mean SSH of a TC-rich year may not reflect the “ocean memory” of TCs for that year. Rather, there should be a time lag. As the authors noted in the introduction, it is difficult to pick out the ocean memory of TCs. I recommend providing more robust rationales for the methods used to analyze the observational data.

3. My third concern is the use of relatively short historical forcing for the ocean simulations. Model drift can affect ocean heat content and potentially obscure the conclusions regarding the impact of TCs. Diagnostics of the model drift for both the spin-up and the TCWIND and FILT runs are needed, particularly for the 0.1 degree experiments. This could be shown with a time series of subsurface ocean temperatures.

Since the historical simulations are relatively short, I wonder how climate variability might influence the mean state in both TCWIND and FILT runs and the Kuroshio current response. For example, a strong multi-year El Niño event could lead to an "El Niño-like" mean state, which could lead to very different Kuroshio currents and heat transport than a "La Niña-like" mean state, possibly exaggerating the impact of TCs. It would be useful to compare the Niño 3.4 index and the mean state large-scale wind and temperature patterns between TCWIND and FILT (similar to the analysis in Extended Figure 9), and to discuss how much of the mean state difference could be influenced by climate variability. Similarly, when analyzing the observational data, the difference between TC-rich and TC-poor years is likely directly connected with ENSO, which makes it difficult to separate the impact of TCs. This is an important point and I suggest adding this as a caveat in the conclusions.

Other comments:

1. Line 824: Add units for SSH anomaly.
2. Extended Figure 3, the structure of the SSH anomalies in the observations and the model are different. The model shows an SSH shoaling to the right of the section center, while AVISO and HYCOM show maximum shoaling at the center. Do the authors have plausible explanations for the difference?
3. Line 861: The order of legends in the caption is incorrect. Brown is anticyclonic eddies in the figure.
4. Line 887 Extended Figure 10: What does the red contour represent?
5. Line 895 Extended Figure 11d: Are these observational anomalies or in the models?

Reviewer #2

(Remarks to the Author)

Review for "Oceanic memory of tropical cyclones moderates the 1 Kuroshio current "

This manuscript investigates the oceanic response to tropical cyclones in the Kuroshio region, focusing on the role of oceanic memory in the response to short-term atmospheric perturbations, tropical cyclones. The study highlights the potential impact of these perturbations on large-scale ocean circulation. While the work provides valuable insights into the influence of tropical cyclones on the western boundary current, the phenomena addressed in the manuscript are highly localized. Moreover, the discussion and evidence linking the findings to large-scale circulation processes are insufficient. Additionally, the core idea of the manuscript—cumulative effects of short-term atmospheric forcing on the ocean—lacks sufficient novelty to distinguish it within the broader context of current research.

Major comments:

A prior study has addressed a closely related topic: the impact of tropical cyclones on the Kuroshio (reference details provided below). This study demonstrated that the effects of tropical cyclones passing over the Kuroshio region, which typically last less than a few days, can persist for about a month, influencing the intensity of the Kuroshio over the same oceanic response period. Furthermore, it provided a detailed analysis of the dynamics by which tropical cyclones affect the subsurface thermal structure, presenting these impacts in a quantitatively robust manner. The two primary topics of the submitted manuscript—ocean memory and subsurface responses—align closely with the findings of the aforementioned study. Consequently, these results are difficult to consider as novel contributions to the field. Park, J. H., Pak, G., Kim, E. J., & Kang, S. K. (2021). Impact of tropical cyclones on geostrophic velocity of the western boundary current. *Geophysical Research Letters*, 48(18), e2021GL094355.

Reviewer #3

(Remarks to the Author)

Review of:

Oceanic memory of tropical cyclones moderates the Kuroshio current

By:

Z Zhang et al

Submitted to:

The study analyzes the impacts of the tropical cyclones on the kuroshio current using observations and high-resolution models. The authors find that TC-induced mixing and thermal responses leads to differential responses in the kuroshio current, and the cumulative effect can have important climate implications.

The findings are interesting and noteworthy, however much of the discussion is technical and directed to more specialized audiences in physical oceanography and related fields.

The paper could be publishable in Nature Communications, but would require major changes to improve the reporting/discussion of results and connections to climate. Otherwise i recommend submitting to a more specialized physical oceanography journal.

General comments:

The paper is a strong model paper, but it needs more quantitative discussion about the results and large-scale contributions, such as processes influencing western boundary current variability. Much of the biggest impacts on ocean heat transport are buried at the end and in the last figure.

Also include more quantitative review of past literature focusing on TC-ocean connections.

The model sensitivity analysis focuses on winds only. What about other contributions such as surface fluxes and precipitation?

How significant are the TC results compared to background current variability, including intensity, and spatial variations during TC seasons when no TCs are occurring?

Specific comments:

L41-44: These results lack perspective. I suggest including some broader impacts. On what time scales are these effects? What is the cumulative effect on ocean heat transport or mixing budgets?

L52: Remove "Besides"

L56: Anomalies of what, ocean heat content?

L59: "memories" is vague. What is the physical quantity?

L61: The broader scale climate connections are vague. I suggest being more specific early in the paper (abstract/intro paragraphs) highlighting the importance of the result for climate.

L69-72: The statement about the large-scale ocean is somewhat vague, It might be helpful to add more discussion about relative contributions of upper and lower transports, seasonal variability of surface currents, in order to help add perspective about the importance of TCs.

L84: Terminology such as "baroclinic geostrophic" lacks context and needs broader context in terms of climate processes.

L88: What is a "barotropic trough"?

L93-96: These different terms need to be introduced and it would be helpful to highlight/quantify the relative contributions of each to ocean heat transport and/or vertical mixing budgets.

L180-182: This is a nice example of quantifying the impact, but it would be helpful to frame in terms of broader and larger-scale quantities? What is the net effect?

L241-243: The result linking cumulative effects of TCs density fields around the kuroshiro is very interesting and relevant. I suggest

L253-255: Is there a seasonal effect on steering flow? Have you considered looking at early versus later season storms?

L308-310: Nice use of a lower resolution model to cut effects of mesoscale eddies, but what about resolution dependencies in the model?

L342-343: Need numbers here from previous studies or at least first order estimates of ocean heat content, transport, etc.

Version 1:

Reviewer comments:

Reviewer #1

(Remarks to the Author)

I appreciate the author's efforts in revising this manuscript. My major comments in the last round of review have been sufficiently addressed, and the revised manuscript has been improved.

I agree with other reviewers that this study is a bit lacking in its novelty and impact, and may be more suitable for JGR-

Oceans. I will defer to the Editor's judgment on whether this manuscript is suitable for Nature Communications.

Reviewer #3

(Remarks to the Author)

The authors addressed all my comments. I recommend publication.

Point-to-point responses to the reviewers' comments

On

“Oceanic memory of tropical cyclones moderates the Kuroshio current”

We wish to thank the Editor and three anonymous reviewers for their valuable time, careful evaluation, and constructive comments. We have taken all comments seriously and have modified the manuscript accordingly. Following are our point-to-point responses, where the comments are in **bold** and *italic*, our responses are in plain print, and the revised text in the manuscript is indicated in blue.

Responses to Reviewer #1

General Overview:

This manuscript investigates the influence of “long-term” ocean memory of TCs on the Kuroshio current. The authors conduct two sets of ocean model simulations, forced with and without TC winds, at both eddy-resolving and non-eddy-resolving resolutions, to isolate the impact of TC winds. In addition, they analyze TCs’ impact using satellite observations and HYCOM reanalysis through composite analysis. The authors conclude that Northwestern Pacific TCs can induce ocean temperature anomalies in the Kuroshio region, which modulate the Kuroshio current through geostrophic processes.

This is a very interesting and comprehensive study and it has the potential to make an important contribution to the field. The manuscript is well-written, and the results are generally robust. However, I have several major concerns, particularly related to the methodologies for analyzing reanalysis and satellite observations. I am concerned that these methods do not sufficiently isolate the impact of TCs. Other concerns relate to the role of large-scale modes of climate variability and model drift. I recommend major revision and hope my suggestions can help improve the manuscript.

Response: We are very grateful to the reviewer for crucial evaluations and detailed modification suggestions, all of which are essential for us to improve the manuscript. We have strived to enhance the robustness of our methodology. To better isolate the impacts of tropical cyclones (TCs), more in-depth composite analysis and comparisons between TC-rich and TC-poor years are conducted. The potential influence of large-scale modes of climate variability and model drift are also investigated and discussed. The details are provided below.

Major comments:

1. My primary concern is whether the composite analysis of the observational data effectively separates the “ocean memory” of TCs on the Kuroshio current. Given the rich eddy activities and the strong exchange of heat and mass in this region, it is hard to argue the temporal changes are induced by TCs alone. It is somewhat surprising that a composite analysis can still capture TCs’ influence after 120-180 days, as shown in Figure 3 and the Extended Figures 3-5.

For example, in Extended Figure 3, the SSH changes in AVISO and HYCOM are

most significant in the first 30 days following the TC passage and have a clear trend of rightward propagation. After 30 days, the SSH shoaling is confined to the section center and is mostly stationary. These are likely two distinct processes: the first involves a short-term response to TCs, and the second is likely driven by longer timescales and may not necessarily be a “memory” of TCs. Moreover, the composite analysis suggests that during the 180 days, only TCs near the section can impact this region and that their impact is confined locally. However, other processes could also affect the boundary current, and TC-induced ocean heat anomalies farther away from the region could be transported to the Kuroshio by ocean currents.

I recommend providing more details on how the seasonal cycles and trends are removed, as well as any post-processing applied to the data. It also would be helpful to extend the composite temporal analysis (i.e., in Extended Figure 3) beyond 180 days to identify if there’s any signal of repeating seasonal cycles or trends. In addition, I would also suggest performing the same composite temporal analysis on the TCWIND experiment and comparing the results with the anomalies calculated as the difference between TCWIND and FILT.

Response: We thank the reviewer for raising these critical points. Following the comment, we supplemented the data processing with more details, extended the temporal analysis, and conducted the same composite temporal analysis on the TCWIND experiment to compare the anomalies between the two experiments. To our understanding, the more pronounced SSH response in the first 30 days could be related to the recovery stage of cold wakes (typically within one month) while the smaller SSH response afterward is mostly due to the long-term subsurface cooling. The possible influences of other processes, such as TC-induced ocean heat anomalies farther away from the region are discussed in Lines 278-282, Page 14-15. Detailed replies are given below.

“Besides, the warm anomalies originating from TCs away from the Kuroshio region can also contribute to the current variation. The anomalies induced by TCs can migrate westward with the background circulation and accumulate near the western boundary current, ultimately leading to a warmer anomaly on the offshore side of the Kuroshio.”

a. Data processing

We have supplemented a subsection of “Composite analysis” in the Method section to introduce the data processing in detail (Lines 668-674, Pages 35), reproduced as:

“Composite analysis is employed to isolate the oceanic responses to TCs for both observations and model data. Before the composite analysis, the seasonal cycle and long-term trend are removed for each grid point. The seasonal cycle is represented by a 365-daily climatology constructed by averaging all data on each day since 1993. A linear trend of each day is obtained via linear regression. By removing these two components, a dataset without seasonal cycle and long-term linear trend is obtained and used to do the composite.”

b. Extended temporal analysis

The composite temporal analysis is extended from day -60 to day +300 relative to TC passage, covering a full annual period. The Extended Fig. 3 is redrawn correspondingly. Results reveal that the seasonal signals or trends are basically removed and do not impact our analysis. In addition, a comparison between TCWIND and FILT reflects a long-term TC-induced change. Overall, consistent results are obtained from AVISO, HYCOM, and TCWIND. Their quantitative differences may arise from artificial smoothing in satellite data (Lu et al. 2023a,b), or underestimations of TC intensities in HYCOM (Hodges et al. 2017). Relevant discussions are reproduced as follows (Lines 674-680, Page 35).

“The composite was performed from -60 days to +300 days relative to TC passage. The TC-induced oceanic changes were obtained by subtracting the mean pre-storm conditions averaged between day -30 and day -3 before TC passage. The results of composite SSHA change (Extended Data Fig. 3) exhibit pronounced SSHA changes following the TC passage, except in the FILT experiment. Moreover, none of the four datasets displays the signals of seasonal cycles or trends, indicating that the composite analysis is capable of isolating TC-induced changes.”

Extended Data Fig. 3. Temporal evolution of composite SSHA (cm) at section 7 associated with the passage of TCs. (a) The tracks and intensity (maximum wind speed) of TCs within 500 km of the section center. Temporal evolution of SSHA at section 7 before and after the TC passage from (b) AVISO, (c) HYCOM, (d) the TCWIND experiment, and (e) the FILT experiment. The linear trend and seasonal cycle have been removed.

c. Composite analysis of TCWIND

As suggested, we have conducted an additional composite analysis of long-term TC-induced changes using the TCWIND experiment and compared it with the difference between TCWIND and FILT, as well as the same composite analysis of FILT (Fig. R1). Results show that the TC-induced changes from the composite temporal analysis of the TCWIND are consistent with those from the anomalies calculated as the difference between TCWIND and FILT. TCs are consistently shown to cause pronounced deep-layer cooling in the Kuroshio core region and upper-layer warming on the offshore side, accompanied by significant deceleration in the central area and acceleration on the offshore side. However, the FILT experiment does not exhibit such characteristics. These results confirm the TC-induced long-term effects on the Kuroshio current.

Fig. R1. Vertical structure of TC-induced potential temperature changes (shading; $^{\circ}\text{C}$) and velocity changes (contours; cm/s) across section 7. (a) Climatology difference between TCWIND and FILT experiment. (b) Tracked mean changes over a period of 120 to 180 days after TC passage relative to the pre-storm state in the TCWIND experiment. (c) Same as (b), but for the FILT experiment. Note that potential temperature anomalies in (b) and (c) are doubled for visualization.

2. Another issue with the observational analysis methodology is the comparison between TC-rich and TC-poor years (Line 148). In Northwestern Pacific, the TC season peaks in late summer to fall, and the annual mean SSH of a TC-rich year may not reflect the “ocean memory” of TCs for that year. Rather, there should be a time lag. As the authors noted in the introduction, it is difficult to pick out the ocean memory of TCs. I recommend providing more robust rationales for the methods used to analyze the observational data.

Response: This is a great point, and we agree. To address this concern, we first

employed a lead-lagged regression as well as correlation analyses between TC intensity and SSHA across the TC season and subsequent seasons. The results revealed that the correlation between TC activities and SSHA within the Kuroshio Current axis core region is negative during TC seasons and positive in subsequent seasons (Extended Data Fig. 11), implying a lagged influence of TCs, as suggested by this comment. Accordingly, the annual averaging period is modified to the average from July of the current year to June of the following year. This change is only related to the SSHA in the revised Fig. 1b, which shows similar and consistent features compared with the original Fig. 1b. A subsection of “Comparison between TC-rich and TC-poor years” and corresponding Extended Fig. 10 and 11 are supplemented in the Method section to better justify the method. The text in the manuscript is reproduced below (Lines 646-665, Pages 34-35).

“The seasonal cycle (subtracting a daily climatology) and linear trend (subtracting a linear trend at each day) were first removed, and then the leading and lagging effects of ENSO were removed via linear regression^{63,64}:

$$SSHA_{NOENSO}(t) = SSHA(t) - \sum_{k=-12}^{12} b_k ONI(t-k)$$

where t is the month in the time series, $SSHA(t)$ is the SSHA without seasonal cycle and linear trend, $SSHA_{NOENSO}$ is SSHA without ENSO effect, k is the lag (negative means lead) month relative to t , and b_k is the regression coefficient on the k th ONI. This approach eliminates the significant correlations between SSHA and ONI within leads and lags of up to 12 months (Extended Data Fig. 10), thereby mitigating ENSO-related influences.

After removing linear ENSO effects, we found that a significant negative correlation exists between SSHA and TC intensity along the Kuroshio axis during the TC season (J-A-S). Because of the recovery of TC-induced upper ocean cooling and a net increase in upper-ocean heat content, a pronounced positive correlation dominates the post-TC season, highlighting the critical role of TC-driven subsurface warming in modulating SSHA variability (Extended Data Fig. 11). These results demonstrate that a comparison of TC-rich and TC-poor years is capable of capturing the impacts of TCs. When calculating the SSHA difference, the annual averaging period was set from July of the current year to June of the following year, accounting for the time lag of the oceanic memory of TC.”

Finally, we note here that comparing TC-rich and TC-poor years is one of several ways to demonstrate TCs’ impacts, given the difficulty of isolating TCs’ impact based on observations. Several other evidences are presented in this study, such as the composite analysis before and after TCs’ passage using altimetry, indicating the robustness of the results.

3. My third concern is the use of relatively short historical forcing for the ocean simulations. Model drift can affect ocean heat content and potentially obscure the conclusions regarding the impact of TCs. Diagnostics of the model drift for both the

spin-up and the TCWIND and FILT runs are needed, particularly for the 0.1 degree experiments. This could be shown with a time series of subsurface ocean temperatures. Since the historical simulations are relatively short, I wonder how climate variability might influence the mean state in both TCWIND and FILT runs and the Kuroshio current response. For example, a strong multi-year El Niño event could lead to an "El Niño-like" mean state, which could lead to very different Kuroshio currents and heat transport than a "La Niña-like" mean state, possibly exaggerating the impact of TCs. It would be useful to compare the Niño 3.4 index and the mean state large-scale wind and temperature patterns between TCWIND and FILT (similar to the analysis in Extended Figure 9), and to discuss how much of the mean state difference could be influenced by climate variability.

Similarly, when analyzing the observational data, the difference between TC-rich and TC-poor years is likely directly connected with ENSO, which makes it difficult to separate the impact of TCs. This is an important point and I suggest adding this as a caveat in the conclusions.

Response: We appreciate the insightful and detailed suggestions. Following the comment, the time series of subsurface ocean temperatures are examined, the influence of climate variability is investigated, and a caveat is added to the conclusions as suggested.

a. Model drift

The time series of subsurface temperatures are drawn below as Fig. R2 to examine the possible model drift issue. There is no significant difference between the two experiments. A relevant discussion is added in Lines 772-775, Page 40:

“The global mean potential temperature reaches a near-equilibrium state after approximately two decades of spin-up (not shown), with no noticeable difference in model drift between the two experiments during the historical simulations.”

Fig. R2. Time evolution of global potential temperature averaged over the upper 2000 m for spin-up and experiments.

b. Climate variability

To investigate how much climate variability might influence the mean state in both TCWIND and FILT runs, we analyzed the ENSO characteristics to investigate the possible influence of climate variability, shown in the newly added Fig. 6 and Extended Data Fig. 2, including (1) comparisons of Niño 3.4 index (referred to the Oceanic Niño Index in the manuscript; ONI), (2) temperature and velocity differences between two experiments during El Niño and La Niña years, respectively, and (3) inter-ENSO phase differences (El Niño minus La Niña) within each experiment. Results show nearly identical ENSO states among observations, TCWIND, and FILT experiments (Extended Data Fig. 2), with very similar large-scale mean state differences (Fig. R3). From the newly added Fig. 6 in the manuscript, the TC influence is shown to be more pronounced during El Niño years than La Niña years but the spatial pattern is consistent, indicating the robustness of the TCs' impact.

Extended Data Fig. 2. Time series of Oceanic Niño Index (ONI) and TC intensity in the Kuroshio region. Oceanic Niño Index (ONI) is defined as a 3-month running mean of SST anomalies in the Niño 3.4 region (5°S - 5°N , 120° - 170°W). TC intensity, defined as annual accumulated wind power input near the Kuroshio area (110° to 140°E , 15° to 35°N), is calculated at 1-hour intervals using linearly interpolated position and maximum wind speed from the 6-hourly IBTrACs best-track archive. The dashed yellow line shows the average intensity from 1993 to 2019. The TC-rich and TC-poor years are marked by red plus and blue circles, respectively.

Fig. R3. ENSO-related large-scale wind stress difference in (a) TCWIND experiment and (b) FILT experiment calculated as the differences between El Niño years and La Niña years. Similar to (a) and (b), the subsurface temperature anomalies (averaged over 50-100 m) related to ENSO in (c) TCWIND experiment and (d) FILT experiment are shown.

c. Caveat addition

We agree that the difference between TC-rich and TC-poor years is likely affected by ENSO variability. In the modified manuscript, a linear regression approach is used to largely remove the ENSO’s impact. Other approaches are used in our study to strengthen the evidence, such as the composite analysis before and after TCs’ passage using altimetry, reanalysis and TCWIND simulation. Multiple lines of evidence suggest the robustness of our conclusion. We also followed the comment to add a caveat related to the interference with ENSO (Lines 154-161, Page 8; Line 435-439, Page 23):

“One limitation of our TC-rich vs TC-poor composite approach is that climate variability related to ENSO, which influences Pacific TC activity, could masquerade as an apparent Kuroshio current effect. We guard against this not only by our use of linear regression to remove and estimate ENSO-related impacts (see Methods), but by in addition performing separate composites for El Niño years and La Niña years. The results of these analyses demonstrate that the TCs impacts is robust with respect to ENSO influences (this matter is revisited later).”

“In addition, however, TC activity in the Northwest Pacific is strongly regulated by ENSO, where TCs are more active, stronger, and southward-shifted during El Niño years^{59,60}. Correspondingly, the TC-induced changes in the Kuroshio are stronger during El Niño years than during La Niña years, with a similar spatial domain of influence (Figs. 6a-c).”

Figure 6. ENSO-related anomalies in potential temperature (50-100 m; shading) and current velocity (0-50 m; vectors) during October–December. Difference between TCWIND and FILT for (a) all years (1993-2016), (b) El Niño years, and (c) La Niña years. Differences between El Niño and La Niña years for (d) HYCOM reanalysis, (e) TCWIND experiment, and (f) FILT experiment.

Other comments:

1. Line 824: Add units for SSH anomaly.

Response: Thanks. The unit of SSH anomaly “(cm)” has been added in the revised Extended Data Fig. 3 and its caption on Line 953, Page 50.

2. Extended Figure 3, the structure of the SSH anomalies in the observations and the model are different. The model shows an SSH shoaling to the right of the section center, while AVISO and HYCOM show maximum shoaling at the center. Do the authors have plausible explanations for the difference?

Response: Thanks. We propose that the primary cause is the slight difference (~10 km) in the Kuroshio's positions across these datasets. In the composite analysis of Extended Fig. 3, the section is fixed and is derived from the TCWIND experiment, thus leading to a slight bias in the center of the SSHA bias. We additionally conducted composite analyses using the same methodology as in Extended Fig. 3, except that the composite centers and sections were derived separately for each dataset. Figure R4 presents the corresponding post-TC SSHA changes, which resemble those in Fig. 1c of the main manuscript and yield essentially the same results.

Fig. R4. Tracked mean SSHA changes across the section based on the flow axis. The anomaly is averaged over a period of 120 to 180 days after TC passage relative to the pre-storm state. Shading indicates standard errors which are calculated as the standard deviation divided by the square root of the sample size. The black, red, and blue curves are obtained from the TCWIND experiment, AVISO, and HYCOM reanalysis, respectively.

3. Line 861: *The order of legends in the caption is incorrect. Brown is anticyclonic eddies in the figure.*

Response: Thanks for pointing this out. The order has been corrected (Line 992, Page 54). We have re-checked all the captions carefully to avoid such careless mistakes.

4. Line 887 Extended Figure 10: *What does the red contour represent?*

Response: Thanks. The red contours denote the 10-cm/s threshold of simulated current velocity in the TCWIND experiment. This information is clarified as “with red contours denoting the 10 cm/s threshold” in Line 1044, Page 59.

5. Line 895 Extended Figure 11d: *Are these observational anomalies or in the models?*

Response: We appreciate this careful check. Both observations and models are incorporated in this figure (the revised Extended Fig. 14), with color shading and red dashed contours denoting SST anomalies from TCWIND experiment and observations, respectively. This information is clarified in Lines 1055-1057, Pages 60-61.

“The red contours are derived from observations while the color shading represents results in the TCWIND experiment.”

References

- Hodges, K., A. Cobb, and P. L. Vidale, 2017: How Well Are Tropical Cyclones Represented in Reanalysis Datasets? *J. Clim.*, **30**, 5243–5264, <https://doi.org/10.1175/JCLI-D-16-0557.1>.
- Lu, Z., G. Wang, and X. Shang, 2023a: Observable Large-Scale Impacts of Tropical

Cyclones on the Subtropical Gyre. *Journal of Physical Oceanography*, **53**, 2189–2209, <https://doi.org/10.1175/JPO-D-22-0230.1>.

—, —, —, and X. Xie, 2023b: Uncertainties in altimetry observations of eddy changes induced by tropical cyclones. *J. Phys. Oceanogr.*, **53**, 113–129, <https://doi.org/10.1175/JPO-D-22-0115.1>.

Responses to Reviewer #2

General Overview:

This manuscript investigates the oceanic response to tropical cyclones in the Kuroshio region, focusing on the role of oceanic memory in the response to short-term atmospheric perturbations, tropical cyclones. The study highlights the potential impact of these perturbations on large-scale ocean circulation. While the work provides valuable insights into the influence of tropical cyclones on the western boundary current, the phenomena addressed in the manuscript are highly localized. Moreover, the discussion and evidence linking the findings to large-scale circulation processes are insufficient.

Additionally, the core idea of the manuscript—cumulative effects of short-term atmospheric forcing on the ocean—lacks sufficient novelty to distinguish it within the broader context of current research.

Response: We appreciate the feedback provided by the reviewer. We hope to better explain the novelty of our study here; and in the revised manuscript we have streamlined the story and better highlight the broad impacts of tropical cyclones (TCs). We hope you could re-evaluate the study with our explanations:

- (1) Although the Kuroshio current is a localized phenomenon, it is an essential component of the global climate system [see a review by Hu et al. (2015)]. Correspondingly, the long-lasting modulation of Kuroshio current by TCs has profound climate implications.
- (2) This work focuses on the Kuroshio current, which is a crucial part of the large-scale circulation. We also followed this comment to discuss more relevant climate implications of this work (Lines 419-452, Pages 22-24).
- (3) We believe that the cumulative effect of TCs is a novel result. As discussed in the introduction, the present literature is focused very much on the near-real-time effects of individual TCs on the Kuroshio (Ezer 2018; Ezer et al. 2017; Park et al. 2021; Todd et al. 2018; Wei et al. 2014) or the climate impact on the Kuroshio by TC-induced changes in mesoscale oceanic eddies (Zhang et al. 2020; Jeon et al. 2022). The findings in this work can thus contribute to the community in a long-term and cumulative perspective.

Major comments:

A prior study has addressed a closely related topic: the impact of tropical cyclones on the Kuroshio (reference details provided below). This study demonstrated that the effects of tropical cyclones passing over the Kuroshio region, which typically last less than a few days, can persist for about a month, influencing the intensity of the Kuroshio over the same oceanic response period. Furthermore, it provided a detailed analysis of the dynamics by which tropical cyclones affect the subsurface thermal structure, presenting these impacts in a quantitatively robust manner.

The two primary topics of the submitted manuscript—ocean memory and subsurface responses—align closely with the findings of the aforementioned study. Consequently, these results are difficult to consider as novel contributions to the field.

Park, J. H., Pak, G., Kim, E. J., & Kang, S. K. (2021). Impact of tropical cyclones on geostrophic velocity of the western boundary current. *Geophysical Research Letters*, 48(18), e2021GL094355.

Response: Thanks for the useful suggestion, and we believe these two studies are fundamentally different in objectives and findings. As stated in the introduction section of the manuscript, the work of Park et al. (2021), and some other relevant studies, focused on a near-real time (short-term) effect of TCs. In fact, Park et al. (2021) have already clearly stated that their work is focused on a short-term period over a time scale from days to one month: “we focus on the direct impact of individual TCs on the Kuroshio velocity—an impact which was maintained for ~30 days”, and thus cannot be interpreted as “on a much longer (decadal) time scale”. In this work, we highlight the long-term impacts of cumulative TCs, at time scales of several months or longer. Besides, the methods and results of these two studies are substantially different.

We realize that this misunderstanding could also arise from our unclear presentation in the paper, thanks for raising this. So we have modified the manuscript to make it clearer and more concise. We appreciate the reviewer’s evaluations.

References

- Ezer, T., 2018: On the interaction between a hurricane, the Gulf Stream and coastal sea level. *Ocean Dyn.*, **68**, 1259–1272, <https://doi.org/10.1007/s10236-018-1193-1>.
- , L. P. Atkinson, and R. Tuleya, 2017: Observations and operational model simulations reveal the impact of Hurricane Matthew (2016) on the Gulf Stream and coastal sea level. *Dynam. Atmos. Oceans*, **80**, 124–138, <https://doi.org/10.1016/j.dynatmoce.2017.10.006>.
- Hu, D., and Coauthors, 2015: Pacific western boundary currents and their roles in climate. *Nature*, **522**, 299–308, <https://doi.org/10.1038/nature14504>.
- Jeon, C., D. R. Watts, H. S. Min, D. G. Kim, S. K. Kang, I.-J. Moon, and J.-H. Park, 2022: Weakening of the Kuroshio upstream by cyclonic cold eddies enhanced by the consecutive passages of Typhoons Danas, Wipha, and Francisco (2013). *Front. Mar. Sci.*, **9**, 884768, <https://doi.org/10.3389/fmars.2022.884768>.
- Park, J., G. Pak, E. J. Kim, and S. K. Kang, 2021: Impact of tropical cyclones on geostrophic velocity of the Western Boundary Current. *Geophys. Res. Lett.*, **48**, <https://doi.org/10.1029/2021GL094355>.
- Todd, R. E., T. G. Asher, J. Heiderich, J. M. Bane, and R. A. Luettich, 2018: Transient response of the Gulf Stream to multiple hurricanes in 2017. *Geophys. Res. Lett.*, **45**, 10,509–10,519, <https://doi.org/10.1029/2018gl079180>.
- Wei, J., X. Liu, and D.-X. Wang, 2014: Dynamic and thermal responses of the Kuroshio to typhoon Megi (2004). *Geophys. Res. Lett.*, **41**, 8495–8502, <https://doi.org/10.1002/2014gl061706>.

Zhang, Y., Z. Zhang, D. Chen, B. Qiu, and W. Wang, 2020: Strengthening of the Kuroshio current by intensifying tropical cyclones. *Science*, **368**, 988–993, <https://doi.org/10.1126/science.aax5758>.

Responses to Reviewer #3

General Overview:

The study analyzes the impacts of the tropical cyclones on the kuroshio current using observations and high-resolution models. The authors find that TC-induced mixing and thermal responses leads to differential responses in the kuroshio current, and the cumulative effect can have important climate implications.

The findings are interesting and noteworthy. However, much of the discussion is technical and directed to more specialized audiences in physical oceanography and related fields.

The paper could be publishable in Nature Communications, but would require major changes to improve the reporting/discussion of results and connections to climate. Otherwise I recommend submitting to a more specialized physical oceanography journal.

Response: We thank the reviewer for encouraging evaluation and critical comments toward a stronger manuscript. We have strived to follow all the suggestions to 1) make the paper less technical and 2) strengthen the linkage between the findings and climate variability to put our new findings in context. Detailed comments are shown below.

Major comments:

1. The paper is a strong model paper, but it needs more quantitative discussion about the results and large-scale contributions, such as processes influencing western boundary current variability. Much of the biggest impacts on ocean heat transport are buried at the end and in the last figure.

Response: We appreciate the suggestion, which is a great point that helped us take a broader view of our work. The paper synthesizes observations (altimetry, SST, Argo, TC tracks), reanalysis data (HYCOM) and high-resolution model simulations to provide multi-lines-of-evidence for TCs' impacts on Kuroshio. We intend to start from the observed and simulated impacts, to explain the mechanism and then extend to large-scale impacts (volume and heat transport). While the general storyline is clear, we agree that it might not be streamlined well, so we have tried to improve the paper writing in the revised manuscript and hope to make it more easy-to-follow for non-specialists. For example, we have supplemented more reviews on TC-ocean connections (see Major Comments #2) to provide a general view of TC's roles in shaping the climate. We have also added some metrics of the Kuroshio to ground discussion of TC-modulated changes (see Specific Comments #6).

Following the comments, the abstract, introduction, and discussion have been carefully modified to strengthen the quantitative discussion and its linkage with large-scale contributions. Some of them are quoted below. In the abstract: “Consequently, the net effect of TCs is to reduce the Kuroshio’s meridional heat transport by \$0.02\pm 0.02\$ PW. On seasonal and interannual scales, TC-induced changes are also comparable to the background Kuroshio variability, implying the TCs’ long-term accumulated impacts on ocean circulations and climate.” (Lines 43-47, Page 2). In the conclusion and discussion: “The inter-annual variation of the Kuroshio itself is closely linked to ENSO

through westward-propagated baroclinic Rossby waves as well as mesoscale eddies in the subtropical countercurrent (STCC) region^{56–58}.” (Lines 432-435, Page 23)

2. Also include more quantitative review of past literature focusing on TC-ocean connections.

Response: Thanks for the suggestion. Following the comment, we have supplemented more quantitative review of TC-ocean connections (e.g., Lines 64-69, Page 3). TC-related changes in ocean heat content and the possible climate consequences are discussed on Lines 352-353, Page 19 and Lines 355-357, Page 19. A few changes are provided below.

“These long-lasting temperature anomalies have profound impacts on global climate patterns. For example, the ocean heat pump is suggested to account for approximately 15% of peak global ocean heat transport¹⁴; the re-emergence of subsurface warm anomalies within the seasonal thermocline may reduce the SST seasonal cycle by about 10%⁷; and these long-lasting subsurface warm anomalies can enhance the subtropical cell by approximately 3%¹¹.”

“Previous studies have shown that TCs contribute to an overall increase in ocean heat content after the cold wake recovery, but the estimates exhibit considerable uncertainties in previous observations and models (0.14 PW to 1.4 ± 0.7 PW)^{6,7,14,50}. Besides, these warm anomalies are suggested to be transported equatorward and poleward, essential for maintaining global heat balance. However, the magnitude of TCs’ contribution to meridional heat transport also remains uncertain, ranging from 0.035 PW to 0.05 PW from different estimates^{3,11,51}.”

3. The model sensitivity analysis focuses on winds only. What about other contributions such as surface fluxes and precipitation?

Response: Thanks for this caution. The wind, surface fluxes and rainfall of TCs are all incorporated together in the TCWIND experiment. This information is clarified in the revised manuscript (Lines 803-804, Page 41). Since previous studies suggest that surface heat flux during the forcing stage plays a minor role in oceanic response to TCs (e.g., Price 1981; D’Asaro et al. 2007; Zhang 2023), we investigated the contribution of precipitation through an additional model experiment by removing TC rainfall (revised Extended Data Fig. 15). A comparison of model results with and without incorporating TC rainfall reveals that the contribution of TC precipitation is marginal relative to that of TC wind (Extended Data Fig. 16). This additional experiment is briefly introduced by adding a “Filtering TC-related precipitation” subsection in the Method section. Relevant introduction and discussion about the precipitation contribution are reproduced as below (Lines 803-814, Pages 41-42).

“The wind, surface heat flux, and rainfall of TCs are all incorporated together in the TCWIND experiment. To investigate precipitation’s contribution, an additional experiment is conducted by filtering the TC rainfall. Precipitation rates within a 500 km radius of each TC center were replaced with background values averaged from days –5 to –2 and days +2 to +5 relative to TC passage. A linear transition zone between 500 km and 600 km from the TC center was implemented to ensure spatial continuity in the

filtered precipitation field. A validation from Typhoon Vongfong demonstrates that this method effectively removes TC-associated precipitation signals both temporally and spatially (Extended Data Fig. 15). A comparison of the model results with and without TC rainfall indicates that the long-term contribution of TC precipitation is marginal relative to that of TC winds to the Kuroshio region (Extended Data Fig. 16).”

Extended Data Fig. 15. Example for removing TC-related precipitation. (a) Time series of the averaged rain rate within 500 km. Spatial distribution of rain rate in **(b)** the TCWIND experiment and **(c)** the NoPrec experiment on day 0 (2014-10-8T10:30:00Z). The red star marks the position of TC Super Typhoon Vongfong.

Extended Data Fig. 16. The simulated oceanic responses to TC precipitation at a nominal 1° resolution. (a) TC precipitation-induced anomalies of potential temperature (shading) and current velocity (black vectors) averaged over 50-100 m depth. Red contours and vectors represent anomalies related to TC winds. (d) Vertical structure of potential temperature anomalies related to TC precipitation (shading) and TC wind (contours) across section 7. The anomalies associated with TC precipitation are calculated as the mean state difference between the TCWIND experiment and the NoPrec experiment.

4. How significant are the TC results compared to background current variability, including intensity, and spatial variations during TC seasons when no TCs are occurring?

Response: Thanks for the very useful suggestion. We have demonstrated the TCs' impact on the climatological mean state of Kuroshio, including current velocity and heat transport. In the revised manuscript, we have included additional analyses to compare the TC-induced anomalies to the background current variability (Conclusion and Discussion section on Lines 422-454, Pages 22-24):

“In addition to its climatological mean Kuroshio influence, TC impacts are associated with substantial variability in Kuroshio behaviors. TC-induced potential temperature anomalies and current velocity changes are particularly pronounced during TC-active seasons, gradually decaying afterward but maintaining detectable influences on basin-scale circulation (Extended Data Fig. 9). The percentage of TC-related seasonal variation is 33–50% of the seasonal amplitude in potential temperature anomalies (0.5–1°C versus 1.5–2°C) and ~50% of the amplitude in current velocity (5–10 cm/s versus 10–20 cm/s), underscoring the substantial impact of TCs on seasonal variability of the Kuroshio.

TCs are also associated with substantial inter-annual Kuroshio variability. The inter-annual variation of the Kuroshio itself is closely linked to ENSO through westward-propagated baroclinic Rossby waves as well as mesoscale eddies in the subtropical countercurrent (STCC) region^{56–58}. In addition, however, TC activity in the Northwest Pacific is strongly regulated by ENSO, where TCs are more active, stronger, and southward-shifted during El Niño years^{59,60}. Correspondingly, the TC-induced changes in the Kuroshio are stronger during El Niño years than during La Niña years, with a similar spatial domain of influence (Figs. 6a-c). In the TCWIND experiment, a comparison between El Niño and La Niña years reveals that the Kuroshio region exhibits a weakened core and strengthened offshore flow, with a warm anomaly on the east side of Taiwan Island but a cold anomaly on the east side of Luzon Island (Fig. 6e). These features are consistent with reanalysis data (Fig. 6d), but are absent in the FILT experiment (Fig. 6f). Given that the ENSO state is nearly identical in both experiments (Extended Data Fig. 2), we infer that the difference between these two experiments demonstrates the effect of TC activity in causing inter-annual variability in the Kuroshio. These relationships merit further study.”

Figure 6. ENSO-related anomalies in potential temperature (50-100 m; shading) and current velocity (0-50 m; vectors) during October–December. Difference between TCWIND and FILT for (a) all years (1993-2016), (b) El Niño years, and (c) La Niña years. Differences between El Niño and La Niña years for (d) HYCOM reanalysis, (e) TCWIND experiment, and (f) FILT experiment.

Extended Data Fig. 2. Time series of Oceanic Niño Index (ONI) and TC intensity in the Kuroshio region. Oceanic Niño Index (ONI) is defined as a 3-month running mean of SST anomalies in the Niño 3.4 region (5°S-5°N, 120°-170°W). TC intensity, defined as annual accumulated wind power input near the Kuroshio area (110° to 140°E, 15° to 35°N), is calculated at 1-hour intervals using linearly interpolated position and maximum wind speed from the 6-hourly IBTrACs best-track archive. The dashed yellow line shows the average intensity from 1993 to 2019. The TC-rich and TC-poor

years are marked by red plus and blue circles, respectively.

Extended Data Fig. 9. Simulated seasonal cycle of TC-induced potential temperature anomalies (50-100 m; shading) and current velocity anomalies (0-50m; vectors). Panels show anomalies for (a) January to March (J-F-M), (b) April to June (A-M-J), (c) July to September (J-A-S), and (d) October to December (O-N-D). The anomalies are calculated as the difference of the long-term mean climatology (1993-2016) between the TCWIND and FILT experiment for respective seasons.

Specific comments:

1. L41-44: These results lack perspective. I suggest including some broader impacts. On what time scales are these effects? What is the cumulative effect on ocean heat transport or mixing budgets?

Response: Thanks for this suggestion. This sentence has been revised and expanded to include broader impacts(Lines 40-47, Page 2), as below:

“For climatological mean state, the cumulative effects of TCs contribute to the strengthening of the upper right flanks of the Kuroshio current by ~15% while weakening of the Kuroshio current along its main axis by ~4% through geostrophic processes. Consequently, the net effect of TCs is to reduce the Kuroshio’s meridional heat transport by 0.02 ± 0.02 PW. On seasonal and interannual scales, TC-induced changes are also comparable to the background Kuroshio variability, implying the TCs’

long-term accumulated impacts on ocean circulations and climate.”

2. L52: Remove “Besides”

Response: Thanks. “Besides” has been removed on Line 55, Page 3.

3. L56: Anomalies of what, ocean heat content?

Response: Thanks. It refers to temperature anomalies. The “temperature” has been added in Line 60, Page 3 accordingly.

4. L59: “memories” is vague. What is the physical quantity?

Response: Thanks for pointing it out. We have explained this word more explicitly on Lines 62-63, Page 3. “(here “memories” indicate TC-induced anomalies that last longer than its life, such as subsurface temperature anomalies)”

5. L61: The broader scale climate connections are vague. I suggest being more specific early in the paper (abstract/intro paragraphs) highlighting the importance of the result for climate.

Response: Thanks. We have followed the comment to revise both the abstract and introduction to make them more specific to highlight the importance of the result for climate:

“For climatological mean state, the cumulative effects of TCs contribute to the strengthening of the upper right flanks of the Kuroshio current by ~15% while weakening of the Kuroshio current along its main axis by ~4% through geostrophic processes. Consequently, the net effect of TCs is to reduce the Kuroshio’s meridional heat transport by 0.02 ± 0.02 PW. On seasonal and interannual scales, TC-induced changes are also comparable to the background Kuroshio variability, implying the TCs’ long-term accumulated impacts on ocean circulations and climate.” (Lines 40-47, Page 2)

“These long-lasting temperature anomalies have profound impacts on global climate patterns. For example, the ocean heat pump is suggested to account for approximately 15% of peak global ocean heat transport¹⁴; the re-emergence of subsurface warm anomalies within the seasonal thermocline may reduce the SST seasonal cycle by about 10%⁷; and these long-lasting subsurface warm anomalies can enhance the subtropical cell by approximately 3%¹¹.” (Lines 64-69, Page 3)

6. L69-72: The statement about the large-scale ocean is somewhat vague. It might be helpful to add more discussion about relative contributions of upper and lower transports, seasonal variability of surface currents, in order to help add perspective about the importance of TCs.

Response: Thanks for this insightful suggestion. We have added more relevant information about Kuroshio here and provided a basis for discussing TCs’ role in the results section on Lines 77-81, Page 4.

“These currents are typically near-surface-intensified, with current speeds exceeding 0.1 m s^{-1} in the upper 1000 m¹⁸. They exhibit significant seasonal variability;

for instance, the surface Kuroshio displays seasonal current speed fluctuations^{19,20} of about 0.05–0.1 m s⁻¹ and volume transport variations²¹ of approximately 0.35 Sv.”

7. L84: Terminology such as “baroclinic geostrophic” lacks context and needs broader context in terms of climate processes.

Response: Thanks. The sentence has been rewritten through replacing “baroclinic” with “the reduced pressure gradient by density changes^{26,30}” (Lines 95-96, Page 5).

8. L88: What is a “barotropic trough”?

Response: The “barotropic trough” has been replaced by “SSH trough” (Lines 98, Page 5).

9. L93-96: These different terms need to be introduced and it would be helpful to highlight/quantify the relative contributions of each to ocean heat transport and/or vertical mixing budgets.

Response: Thanks. We have tried to follow the suggestion to inspect the contribution of TC-eddy interaction on heat transport in previous studies, while we find that some of these studies have derived contrary results. A conclusion for the relative contributions of mesoscale oceanic eddies has not been supplemented into the manuscript yet.

10. L180-182: This is a nice example of quantifying the impact, but it would be helpful to frame in terms of broader and larger-scale quantities? What is the net effect?

Response: We appreciate the encouraging comment. We have added two subtitles of “TCs’ impact on the Kuroshio current” and “TCs’ impact on the volume transport” to make the motivation clearer. The net effects of TCs are a reduction of Kuroshio’s meridional heat transport by about 0.02±0.02 PW, and a decrease of volume transport by 0.3 Sv. This information is stressed in the abstract and conclusions accordingly (Lines 43-44, Page 2; Lines 411-412, Page 22).

“Consequently, the net effect of TCs is to reduce the Kuroshio’s meridional heat transport by 0.02±0.02 PW.”

“... leading to an overall decrease of volume transport by about 0.3 Sv within 200 km across the Kuroshio.”

11. L241-243: The result linking cumulative effects of TCs density fields around the Kuroshio is very interesting and relevant.

Response: We appreciate the positive evaluation.

12. L253-255: Is there a seasonal effect on steering flow? Have you considered looking at early versus later season storms?

Response: Following the comment, we analyzed seasonal variations in the propagation of TC-induced subsurface temperature anomalies. As illustrated in Fig. R4, these anomalies exhibit consistent leftward propagation relative to TC tracks across different

seasons, with speeds of approximately 0.07 m/s (early season) and 0.09 m/s (later season). These velocities are in agreement with theoretical estimates for the westward group velocity of first-baroclinic-mode Rossby waves (Jansen et al. 2010), which typically accelerate with the decrease in latitude. The slight seasonal differences result from the latitudinal shifts in the distribution of TC activity in different seasons, with TCs in early and later seasons occurring on average at 23°N and 18°N, respectively.

Fig. R4. Temporal evolution of along-track-averaged subsurface temperature (50-100 m) anomalies. All composites encompass all TCs of category 1 to 5 intensity in the North Western Pacific (100° to 180°E; 0 to 40°N) during 1993-2016. The anomalies are calculated relative to the pre-TC passage period (days -30 to -3) with the seasonal cycle removed.

13. L308-310: Nice use of a lower resolution model to cut effects of mesoscale eddies, but what about resolution dependencies in the model?

Response: Thanks for the commendation and insightful question. The coarse-resolution simulations produce weaker and wider Kuroshio current than the fine-resolution simulations (30 cm/s vs 100 cm/s, and 200 km vs 100 km). However, the TC-induced changes in thermal structures and currents are consistent for both fine and coarse resolution experiments, thus the main conclusion of this study is not dependent on model resolution. A brief discussion is added for the influence of coarse resolution (Lines 317-321, Page 17):

“To investigate the possible influence of mesoscale eddies on the Kuroshio, we conduct an additional set of experiments at a lower 1° resolution, which largely eliminates the contributions of mesoscale eddies but retains TC-induced thermal anomalies⁴². The low-resolution model simulates a weaker Kuroshio with a broader flow axis, leading to a less pronounced response to TCs (Fig. 4a).”

14. L342-343: Need numbers here from previous studies or at least first order

estimates of ocean heat content, transport, etc.

Response: Thanks. As suggested, previous estimates using observations and models have been supplemented in Lines 352-353 and 355-357, Page 19.

“the estimates exhibit considerable uncertainties in previous observations and models (0.14 PW to 1.4 ± 0.7 PW).”

“However, the magnitude of TCs’ contribution to meridional heat transport also remains uncertain, ranging from 0.035 PW to 0.05 PW from different estimates^{3,11,51}.”

References

D’Asaro, E., T. B. Sanford, P. P. Niiler, and E. J. Terrill, 2007: Cold wake of Hurricane Frances. *Geophys. Res. Lett.*, **34**, L15609, <https://doi.org/10.1029/2007GL030160>.

Jansen, M. F., R. Ferrari, and T. A. Mooring, 2010: Seasonal versus permanent thermocline warming by tropical cyclones. *Geophys. Res. Lett.*, **37**, 2009GL041808, <https://doi.org/10.1029/2009GL041808>.

Price, J. F., 1981: Upper ocean response to a hurricane. *J. Phys. Oceanogr.*, **11**, 153–175, [https://doi.org/10.1175/1520-0485\(1981\)011<0153:uortah>2.0.co;2](https://doi.org/10.1175/1520-0485(1981)011<0153:uortah>2.0.co;2).

Zhang, H., 2023: Modulation of upper ocean vertical temperature structure and heat content by a fast-moving tropical cyclone. *J. Phys. Oceanogr.*, **53**, 493-508. <https://doi.org/10.1175/JPO-D-22-0132.1>